# FedFed: Feature Distillation against Data Heterogeneity in Federated Learning

**Zhiqin Yang**[1,2*]   **Yonggang Zhang**[2*]   **Yu Zheng**[3]   **Xinmei Tian**[5]

**Hao Peng**[1,6†]   **Tongliang Liu**[4]   **Bo Han**[2]

[1]Beihang University   [2]Hong Kong Baptist University   [3]Chinese University of Hong Kong
[4]Sydney AI Centre, The University of Sydney   [5]University of Science and Technology of China
[6] Kunming University of Science and Technology

## Abstract

Federated learning (FL) typically faces data heterogeneity, i.e., distribution shifting among clients. Sharing clients' information has shown great potentiality in mitigating data heterogeneity, yet incurs a dilemma in preserving privacy and promoting model performance. To alleviate the dilemma, we raise a fundamental question: *Is it possible to share partial features in the data to tackle data heterogeneity?* In this work, we give an affirmative answer to this question by proposing a novel approach called **Fed**erated **Fe**ature **d**istillation (FedFed). Specifically, FedFed partitions data into performance-sensitive features (i.e., greatly contributing to model performance) and performance-robust features (i.e., limitedly contributing to model performance). The performance-sensitive features are globally shared to mitigate data heterogeneity, while the performance-robust features are kept locally. FedFed enables clients to train models over local and shared data. Comprehensive experiments demonstrate the efficacy of FedFed in promoting model performance. The code is publicly available at: https://github.com/tmlr-group/FedFed

## 1   Introduction

Federated learning (FL), beneficial for training over multiple distributed data sources, has recently received increasing attention [1, 2, 3]. In FL, many clients collaboratively train a global model by aggregating gradients (or model parameters) without the need to share local data. However, the major concern is heterogeneity issues [4, 5, 6] caused by Non-IID distribution of distributed data and diverse computing capability across clients. The heterogeneity issues can cause unstable model convergence and degraded prediction accuracy, hindering further FL deployments in practice [1].

To address the heterogeneity challenge, the seminal work, federated averaging (FedAvg), introduces the model aggregation of locally trained models [4]. It addresses the diversity of computing and communication but still faces the issue of client drift induced by data heterogeneity [7]. Therefore, a branch of works for defending data heterogeneity has been explored by devising new learning objectives [6], aggregation strategies [8], and constructing shareable information across clients [9]. Among explorations as aforementioned, sharing clients' information has been considered to be a straightforward and promising approach to mitigate data heterogeneity [9, 10].

However, the dilemma of preserving data privacy and promoting model performance hinders the practical effectiveness of the information-sharing strategy. Specifically, it shows that a limited amount of shared data could significantly improve model performance [9]. Unfortunately, no matter for sharing raw data, synthesized data, logits, or statistical information [10, 11, 12, 7] can

---

*Equal contributions.

†Corresponding author (penghao@act.buaa.edu.cn)

incur privacy concerns [13, 14, 15]. Injecting random noise to data provides provable security for protecting privacy [16, 17]. Yet, the primary concern for applying noise to data lies in performance degradation [18] as the injected noise negatively contributes to model performance. Consequently, effectively fulfilling the role of shared information necessitates addressing the dilemma of data privacy and model performance.

We revisit the purpose of sharing information to alleviate the dilemma in information sharing and performance improvements and ask:

*Q.1 Is it possible to share partial features in the data to mitigate heterogeneity in FL?* This question is fundamental to reaching a new phase for mitigating heterogeneity with the information-sharing strategy. Inspired by the partition strategy of spurious feature and robust feature [19], the privacy and performance dilemma could be solved if the data features were separated into performance-robust and performance-sensitive parts without overlapping, where the performance-robust features contain almost all the information in the data. Namely, the performance-robust features that limitedly contribute to model performance are kept locally. Meanwhile, the performance-sensitive features that could contribute to model generalization are selected to be shared across clients. Accordingly, the server can construct a global dataset using the shared performance-sensitive features, which enables clients to train their models over the local and shared data.

*Q.2 How to divide data into performance-robust features and performance-sensitive features?* The question is inherently related to the spirit of the information bottleneck (IB) method [20]. In IB, ideal features should discard all information in data features except for the minimal sufficient information for generalization [21]. Concerning the information-sharing and performance-improvement dilemma, the features discarded in IB may contain performance-robust features, i.e., private information, thus, they are unnecessary to be shared for heterogeneity mitigation. Meanwhile, the performance-sensitive features contain information for generalization, which are the minimum sufficient information and should be shared across clients for heterogeneity mitigation. Therefore, IB provides an information-theoretic perspective of dividing data features for heterogeneity mitigation in FL.

*Q.3 What if performance-sensitive features contain private information?* This question lies in the fact that a non-trivial information-sharing strategy should contain necessary data information to mitigate the issue of data heterogeneity. That is, the information-sharing strategy unavoidably causes privacy risks. Fortunately, we can follow the conventional style in applying random noise to protect performance-sensitive features because the noise injection approach can provide a de facto standard way for provable security [16]. Notably, applying random noise to performance-sensitive features differs from applying random noise for all data features. More specifically, sharing partial features in the data is more accessible to preserve privacy than sharing complete data features, which is fortunately consistent with our theoretical analysis, see Theorem 3.3 in Sec. 3.3.

**Our Solution.** Built upon the above analysis, we propose a novel framework, named **Fed**erated **Fe**ature **d**istillation (FedFed), to tackle data heterogeneity by generating and sharing performance-sensitive features. According to the question *Q.1*, FedFed introduces a competitive mechanism by decomposing data features $\mathbf{x} \in \mathbb{R}^d$ with dimension $d$ into performance-robust features $\mathbf{x}_r \in \mathbb{R}^d$ and performance-sensitive features $\mathbf{x}_s \in \mathbb{R}^d$, i.e., $\mathbf{x} = \mathbf{x}_r + \mathbf{x}_s$. Then following the question *Q.2*, FedFed generates performance-robust features $\mathbf{x}_r$ in an IB manner for data $\mathbf{x}$. In line with *Q.3*, FedFed enables clients to securely share their protected features $\mathbf{x}_p$ by applying random noise $\mathbf{n}$ to performance sensitive features $\mathbf{x}_s$, i.e., $\mathbf{x}_p = \mathbf{x}_s + \mathbf{n}$, where $\mathbf{n}$ is drawn from a Gaussian distribution $\mathcal{N}(\mathbf{0}, \sigma_s^2\mathbf{I})$ with variance $\sigma_s^2$. To this end, the server can construct a global dataset to tackle data heterogeneity using the protected features $\mathbf{x}_p$, enabling clients to train models over the local private and globally shared data.

We deploy FedFed on four popular FL algorithms, including FedAvg [4], FedProx [6], SCAFFOLD [7], and FedNova [22]. Atop them, we conduct comprehensive experiments on various scenarios regarding different amounts of clients, varying degrees of heterogeneity, and four datasets. Extensive results show that the FedFed achieves considerable performance gains in all settings.

Our contributions are summarized as follows:

1. We pose a foundation question to challenge the necessity of sharing all data features for mitigating heterogeneity in FL with information-sharing strategies. The question sheds light on solving the privacy and performance dilemma, in a way of sharing partial features that contribute to data heterogeneity mitigation (Sec 3.1)

2. To solve the dilemma in information sharing and performance improvements, we propose a new framework FedFed. In FedFed, each client performs feature distillation—partitioning local data into performance-robust and performance-sensitive features (Sec 3.2) —and shares the latter with random noise globally (Sec 3.3). Consequently, FedFed can mitigate data heterogeneity by enabling clients to train their models over the local and shared data.

3. We conduct comprehensive experiments to show that FedFed consistently and significantly enhances the convergence rate and generalization performance of FL models across different scenarios under various datasets (Sec 4.2).

## 2 Preliminary

**Federated Learning.** Federated learning allows multiple clients to collaboratively train a global model parameterized by $\phi$ without exposing clients' data [4, 1]. In general, the global model aims to minimize a global objective function $\mathcal{L}(\phi)$ over all clients' data distributions:

$$\min_{\phi} \mathcal{L}(\phi) = \sum_{k=1}^{K} \lambda_k \mathcal{L}_k(\phi_k), \tag{1}$$

where $K$ represents the total number of clients, $\lambda_k$ is the weight of the $k$-th client. The local objective function $\mathcal{L}_k(\phi_k)$ of client $k$ is defined on the distribution $P(X_k, Y_k)$ with random variables $X_k, Y_k$:

$$\mathcal{L}_k(\phi_k) \triangleq \mathbb{E}_{(\mathbf{x},\mathbf{y}) \sim P(X_k, Y_k)} \ell(\phi_k; \mathbf{x}, \mathbf{y}), \tag{2}$$

where $\mathbf{x}$ is input data with its label $\mathbf{y}$ and $\ell(\cdot)$ stands for the loss function, e.g., cross-entropy loss. Due to the distributed property of clients' data, the global objective function $\mathcal{L}(\phi)$ is optimized round-by-round. Specifically, within the $r$-th communication round, a set of clients $C$ are selected to perform local training on their private data, resulting in $|C|$ optimized models $\{\phi_k^r\}_{k=1}^{|C|}$. These optimized models are then sent to a central server to derive a global model for the $(r+1)$-th communication round by an aggregation mechanism $AGG(\cdot)$, which may vary from different FL algorithms:

$$\phi^{r+1} = AGG(\{\phi_k^r\}_{k=1}^{|C|}). \tag{3}$$

**Differential Privacy.** Differential privacy [16] is a framework to quantify to what extent individual privacy in a dataset is preserved while releasing the data.

**Definition 2.1.** *(Differential Privacy). A randomized mechanism $\mathcal{M}$ provides $(\epsilon, \delta)$-differential privacy (DP) if for any two neighboring datasets $D$ and $D'$ that differ in a single entry, $\forall S \subseteq Range(\mathcal{M})$,*

$$\Pr(\mathcal{M}(D) \in S) \le e^{\epsilon} \cdot \Pr(\mathcal{M}(D') \in S) + \delta.$$

*where $\epsilon$ is the privacy budget and $\delta$ is the failure probability.*

The definition of $(\epsilon, \delta)$-DP shows the difference of two neighboring datasets in the probability that the output of $\mathcal{M}$ falls within an arbitrary set $S$ is related to $\epsilon$ and an error term $\delta$. Similar outputs of $\mathcal{M}$ on $D, D'$ (i.e., smaller $\epsilon$) represent a stronger privacy guarantee. In sum, DP as a de facto standard of quantitative privacy, provides provable security for protecting privacy.

**Information Bottleneck.** Traditionally, in the context of the Information Bottleneck (IB) framework, the goal is to effectively capture the information relevant to the output label $Y$, denoted as $Z$, while simultaneously achieving maximum compression of the input $X$. $Z$ represents the latent embedding, which serves as a compressed and informative representation of $X$, preserving the essential information while minimizing redundancy. This objective can be formulated as:

$$\mathcal{L}_{\text{IB}} = I(X; Y|Z), \quad s.t. \ I(X; Z) \le I_{\text{IB}} \tag{4}$$

where $I(\cdot)$ denote the mutual information and $I_{\text{IB}}$ is a constant. This function is defined as a rate-distortion problem, indicating that IB is to extract the most efficient and informative features.

## 3 Methodology

We detail the Federated Feature distillation (FedFed) proposed to mitigate data heterogeneity in FL. FedFed adopts the information-sharing strategy with the spirits of information bottleneck (IB). Roughly, it shares the minimal sufficient features, while keeping other features at clients[1].

---

[1]Appendix A displays FedFed's overview.

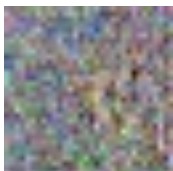 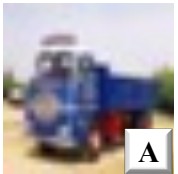 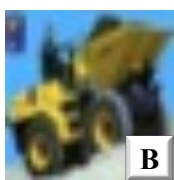 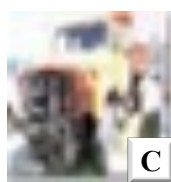

Figure 1: Have a Guessing Game! Question: Which one is the first image from? A or B or C?

## 3.1 Motivation

Data heterogeneity inherently comes from the difference in data distribution among clients. Aggregating models without accessing data would inevitably bring performance degradation. Sharing clients' data benefits model performance greatly, but intrinsically violates privacy. Applying DP to protect shared data looks feasible; however, it is not a free lunch with paying the accuracy loss on the model.

Draw inspiration from the content and style partition of data causes [19], we investigate the dilemma in information sharing and performance improvements through a feature partition perspective. By introducing an appropriate partition, we can share performance-sensitive features while keeping performance-robust features locally. Namely, *performance-sensitive features in the data are all we need for mitigating data heterogeneity!*

We will elucidate the definitions of performance-sensitive features and performance-robust features. Firstly, we provide a precise definition of a valid partition (Definition 3.1), which captures the desirable attributes when partitioning features. Subsequently, adhering to the rules of a valid partition, we formalize the two types of features (Definition 3.2).

**Definition 3.1.** *(Valid Partition). A partition strategy is to partition a variable $X$ into two parts in the same measure space such that $X = X_1 + X_2$. We say the partition strategy is valid if it holds: (i) $H(X_1, X_2|X) = 0$; (ii) $H(X|X_1, X_2) = 0$; (iii) $I(X_1; X_2) = 0$; where $H(\cdot)$ denotes the information entropy and $I(\cdot)$ is the mutual information.*

**Definition 3.2.** *(Performance-sensitive and Performance-robust Features). Let $X = X_s + X_r$ be a valid partition strategy. We say $X_s$ are performance-sensitive features such that $I(X; Y|X_s) = 0$, where $Y$ is the label of $X$. Accordingly, $X_r$ are the performance-robust features.*

Intuitively, performance-sensitive features contain all label information, while performance-robust features contain all information about the data except for the label information. More discussions about Definition 3.1 and Definition 3.2 can be found in Appendix C.1. Now, the challenge turns out to be: Can we derive performance-sensitive features to mitigate data heterogeneity? FedFed provides an affirmative answer from an IB perspective, boosting a simple yet effective data-sharing strategy.

## 3.2 Feature Distillation

Given the motivation above, we propose to distil features such that data features can be partitioned into performance-robust features depicting mostly data and performance-sensitive features favourable to model performance. We design feature distillation and answer Q.2 below.

We draw inspiration from the information bottleneck method [21]. Specifically, only minimal sufficient information is preserved in learned representations while the other features are dismissed [21]. This can be formulated as [2]:

$$\min_{Z} I(X; Y|Z), \ s.t. \ I(X; Z) \le I_{\text{IB}}, \tag{5}$$

where $Z$ represents the desired representation extracted from input $X$ with its label $Y$, $I(\cdot)$ is the mutual information, and $I_{\text{IB}}$ stands for a constant. Namely, given the learned representation $Z$, the mutual information between $X$ and $Y$ is minimized. Meanwhile, $Z$ dismisses most information about the input $X$ so that the mutual information $I(X; Z)$ is less than a constant $I_{\text{IB}}$.

Similarly, in FedFed, only minimal sufficient information is necessary to be shared across clients to mitigate data heterogeneity, while the other features are dismissed before sharing data, i.e., kept locally on the client. The differences between IB and FedFed have two folds: 1) FedFed aims to make

---

[2]We follow the formulation used in Tishby and Zaslavsky [21, 23].

the dismissed features close to the original features (i.e., depict most private data information), while IB focuses on the dissimilarity between the preserved and original features (i.e., contain minimal sufficient information about data); 2) FedFed dismisses information in the data space while IB does that in the representation space. More specifically, the objective of feature distillation is:

$$\min_{Z} I(X; Y | Z), \ s.t. \ I(X; X - Z | Z) \geq I_{\mathsf{FF}}, \tag{6}$$

where $Z$ represents the performance-sensitive features in $X$, $Y$ is the label of $X$, $(X - Z)$ stands for performance-robust features that are unnecessary to be shared, and $I_{\mathsf{FF}}$ is a constant. In Eq. (6), the $(X - Z)$ should represent data mostly conditioned on performance-sensitive features $Z$, while the $Z$ should contain necessary information about the label $Y$. Consequently, learning with the objective can divide data features into performance-sensitive features and performance-robust features, achieving the goal of feature distillation.

To make the feature distillation tractable, we derive an objective equal to the original objective in Eq. (6) (see Appendix C.2 for more details) for client $k$ as follows:

$$\min_{\theta} -\mathbb{E}_{(\mathbf{x}, \mathbf{y}) \sim P(X_k, Y_k)} \log p(\mathbf{y} | \mathbf{z}(\mathbf{x}; \theta)), \ s.t. \ ||\mathbf{z}(\mathbf{x}; \theta)||_2^2 \leq \rho, \tag{7}$$

where $\theta$ is the parameter to be optimized for generating performance-sensitive features $\mathbf{z}(\cdot; \theta)$, $(\mathbf{x}, \mathbf{y})$ represents the input pair drawn from the joint distribution $P(X_k, Y_k)$ of client $k$, $p(\mathbf{y} | \cdot)$ is the probability of predicting $Y = \mathbf{y}$, and $\rho > 0$ stands for a constant. The underlying insight of the objective function in Eq. (7) is intuitive. Specifically, the learned performance-sensitive features $\mathbf{z}(\cdot; \theta)$ are capable of predicting the label $\mathbf{y}$ while having the minimal $\ell_2$-norm.

Unfortunately, the original formulation in Eq. (7) can hardly be used for dividing data features. This is because the feature distillation in Eq. 7 cannot make the preserved features $\mathbf{x} - \mathbf{z}(\mathbf{x}; \theta)$ similar to the raw features $\mathbf{x}$. Namely, the original Eq. (7) cannot guarantee the desired property of the preserved features. To solve the problem, we propose an explicit competition mechanism in the data space. Specifically, we model preserved features, i.e., performance-robust features, with $q(\mathbf{x}; \theta)$ explicitly and model the performance-sensitive features $\mathbf{z}(\mathbf{x}; \theta) \triangleq \mathbf{x} - q(\mathbf{x}; \theta)$ implicitly:

$$\min_{\theta} -\mathbb{E}_{(\mathbf{x}, \mathbf{y}) \sim P(X_k, Y_k)} \log p(\mathbf{y} | \mathbf{x} - q(\mathbf{x}; \theta)), \ s.t. \ ||\mathbf{x} - q(\mathbf{x}; \theta)||_2^2 \leq \rho. \tag{8}$$

Thus, by Eq. 8, performance-sensitive features can predict labels and performance-robust features are almost the same as raw features.

The realization of Eq. (8) is straightforward. To be specific, we can employ a generative model parameterized with $\theta$ to generate performance-robust features, i.e., $q(\mathbf{x}; \theta)$ in Eq. (8). Meanwhile, we train a local classifier $f(\cdot; \mathbf{w}_k)$ parameterized with $\mathbf{w}_k$ for client-$k$ to model the process of predicting labels, i.e., $-\log p(\mathbf{y} | \cdot)$ in Eq. (8). Accordingly, the realization of feature distillation is formulated as follows with $\mathbf{z}(\mathbf{x}; \theta) \triangleq \mathbf{x} - q(\mathbf{x}; \theta)$:

$$\min_{\theta, \mathbf{w}_k} -\mathbb{E}_{(\mathbf{x}, \mathbf{y}) \sim P(X_k, X_k)} \ell(f(\mathbf{z}(\mathbf{x}; \theta); \mathbf{w}_k), \mathbf{y}), \ s.t. ||\mathbf{z}(\mathbf{x}; \theta)||_2^2 \leq \rho, \tag{9}$$

where $\ell(\cdot)$ is the cross-entropy loss, $q(\mathbf{x}; \theta)$ stands for a generative model, and $\rho$ represents a tunable hyper-parameter. Built upon Eq. (9), we can perform feature distillation, namely, the outputs of a generator $q(\mathbf{x}; \theta)$ serve as performance-robust features and $\mathbf{x} - q(\mathbf{x}; \theta)$ are used as performance-sensitive features. We merely share $\mathbf{x} - q(\mathbf{x}; \theta)$ to tackle data heterogeneity by training models over both local and shared data. Algorithm 1 summarizes the procedure of feature distillation.

### 3.3 Protection for Performance-Sensitive Features

Until now, we have intentionally overlooked the overlap between performance-sensitive features and performance-robust features. Since we prioritize data heterogeneity, overlapping is almost unavoidable in practice, i.e., performance-sensitive features containing certain data privacy. Accordingly, merely sharing performance-sensitive features can risk privacy. Thus, we answer Q. 3 below.

**Why employ DP?** The constructed performance-sensitive features may contain individual privacy, thus, the goal of introducing a protection approach is for individual privacy. In addition, the employed protection approach is expected to be robust against privacy attacks [24, 25]. According to the above analysis, differential privacy (DP) is naturally suitable in our scenario of feature distillation. Thus, we employ DP to protect performance-sensitive features before sending them to the server.

---
**Algorithm 1** Feature Distillation
---
**Server input:** communication round $T_d$, DP noise level $\sigma_s^2$
**Client $k$'s input:** local epochs of feature distillation $E_d$, $k$-th local dataset $\mathcal{D}^k$ and rescale to $[0, 1]$
   **Initialization:** server distributes the initial model $\mathbf{w}^0, \theta^0$ to all clients
   **Server Executes:**
   **for** each round $t = 1, 2, \cdots, T_d$ **do**
     server samples a subset of clients $C_t \subseteq \{1, ..., K\}$,
     server **communicates** $\mathbf{w}^t, \theta^t$ to selected clients
     **for** each client $k \in C_t$ **in parallel do**
       $\mathbf{w}_k^{t+1}, \theta_k^{t+1} \leftarrow$ Local_FeatDis $(\mathbf{w}^t, \theta^t, \sigma_s^2)$
     **end for**
     $\mathbf{w}^{t+1}, \theta^{t+1} \leftarrow$ AGG $(\mathbf{w}_k^t, \theta_k^t, k \in C_t)$
   **end for**
   $\mathcal{D}^s = \{\mathcal{D}_k^s\}_{k=1}^K \leftarrow$ Collecting $\mathbf{x}_p$ generated by $k$-th client use Eq (9), where $\mathbf{x}_p = \mathbf{x}_s + \mathbf{n}$

   **Local_FeatDis($\mathbf{w}^t, \theta^t, \sigma_s^2$):**
   **for** each local epoch $e$ with $e = 1, \cdots, E_d$ **do**
     $\mathbf{w}_k^{t+1}, \theta_k^{t+1} \leftarrow$ SGD update use Eq (9).
   **end for**
   **Return** $\mathbf{w}_k^{t+1}, \theta_k^{t+1}$ to server
---

**How to apply DP?** Applying noise (e.g., Gaussian or Laplacian) to performance-sensitive features before sharing can protect features with DP guarantee [26, 16], i.e., $\mathbf{x}_p = \mathbf{x}_s + \mathbf{n}$. Consequently, the server can collect protected features $\mathbf{x}_p$ from clients to construct a global dataset and send the dataset back to clients. To give a vivid illustration, we provide a guessing game[3] in Figure 1, showing 4 images. The first image represents the DP-protected performance-sensitive features of one image (A, B, or C). It is hard to identify which one of the raw images (A, B, C) the first image belongs to.

**Training with DP.** The protected performance-sensitive features are shared, thus, the server can construct a globally shared dataset. Using the global dataset, clients can train local classifier $f(\cdot; \phi_k)$ parameterized by $\phi_k$ with the private and shared data:

$$\min_{\phi_k} \mathbb{E}_{(\mathbf{x},\mathbf{y}) \sim P(X_k, Y_k)} \ell(f(\mathbf{x}; \phi_k), \mathbf{y}) + \mathbb{E}_{(\mathbf{x}_p, \mathbf{y}) \sim P(\mathbf{h}(X_k), Y_k)} \ell(f(\mathbf{x}_p; \phi_k), \mathbf{y}), \tag{10}$$

where $f(\cdot; \phi_k)$ is trained for model aggregation, $(\mathbf{x}_p, \mathbf{y})$ are protected performance-sensitive features that are collected from all clients, and $\mathbf{h}(X_k) := \mathbf{z}(X_k) + n$ with noise $n$. Pseudo-code of how to apply FedFed are listed in Appendix B.

**DP guarantee.** Following DP-SGD [27], we employ the idea of the $\ell_2$-norm clipping relating to the selection of the noise level $\sigma$. Specifically, we realize the constraint in Eq. (9) as follows: $\|\mathbf{z}(\mathbf{x}; \theta)\|_2 \leq \rho \|\mathbf{x}\|$, with $0 < \rho < 1$ (i.e., $\|\mathbf{x}_s\| \leq \rho \|\mathbf{x}\|$). For aligning analyses with conventional DP guarantee, we let the random mechanism on performance-robust features $\mathbf{x}_r$ be $\mathbf{x}_r + n, n \sim \mathcal{N}(0, \sigma_r^2 \mathbf{I})$. Similar for performance-sensitive features, we have $\mathbf{x}_s + n, n \sim \mathcal{N}(0, \sigma_s^2 \mathbf{I})$. Since $\mathbf{x}_r$ is kept locally, an adversary views nothing (or random data without any entropy). This keeps equal to adding a sufficiently large noise, i.e., $\sigma_r = \infty$ on $\mathbf{x}_r$ to make it random enough. Consequently, FedFed can provide a strong privacy guarantee. Moreover, for each client, we show that FedFed requires a relatively small noise level $\sigma$ for achieving identical privacy, which is given in Theorem 3.3 (detailed analysis is left in Appendix D).

**Theorem 3.3.** *Let $\sigma$ be the noise scale for FedFed and $\sigma'$ be the noise scale for sharing raw $\mathbf{x}$. Given identical $\epsilon, \delta$, we attain $\sigma < \sigma'$ such that $\sigma \propto \|\mathbf{x}_s\|^2$.*

Besides each client, we give the privacy analysis for the FL system paired with the proposed FedFed.

**Theorem 3.4** (Composition of FedFed). *For $k$ clients with $(\epsilon, \delta)$-differential privacy, FedFed satisfies $(\hat{\epsilon}_{\hat{\delta}}, 1 - (1 - \hat{\delta})\Pi_i(1 - \delta_i))$-differential privacy,*

---
[3]We reveal the answer in Appendix F.2

Table 1: Top-1 accuracy with(without) FedFed under different heterogeneity degree, local epochs, and clients number on CIFAR-10.

| | centralized training ACC = 95.48% w/(w/o) **FedFed** | | | | | | | |
| --- | --- | --- | --- | --- | --- | --- | --- | --- |
| | ACC↑ | Gain↑ | Round↓ | Speedup↑ | ACC↑ | Gain↑ | Round↓ | Speedup↑ |
| | $\alpha = 0.1, E = 1, K = 10$ (Target ACC =79%) | | | | $\alpha = 0.05, E = 1, K = 10$ (Target ACC =69%) | | | |
| FedAvg | **92.34**(79.35) | 12.99↑ | **39**(284) | ×**7.3**(×1.0) | **90.02**(69.36) | 20.66↑ | **44**(405) | ×**9.2**(×1.0) |
| FedProx | **92.12**(83.06) | 9.06↑ | **62**(192) | ×**4.6**(×1.5) | **90.73**(78.98) | 11.75↑ | **48**(203) | ×**8.4**(×2.0) |
| SCAFFOLD | **89.66**(83.67) | 5.99↑ | **34**(288) | ×**8.4**(×1.0) | **81.04**(37.87) | 43.17↑ | **37**(None) | ×**10.9**(None) |
| FedNova | **92.23**(80.95) | 11.28↑ | **33**(349) | ×**8.6**(×0.8) | **91.21**(65.08) | 26.13↑ | **32**(None) | ×**12.7**(None) |
| | $\alpha = 0.1, E = 5, K = 10$ (Target ACC =85%) | | | | $\alpha = 0.1, E = 1, K = 100$ (Target ACC =49%) | | | |
| FedAvg | **93.24**(83.79) | 9.45 ↑ | **17**(261) | ×**15.4**(×1.0) | **84.06**(49.72) | 34.34↑ | **163**(967) | ×**5.9**(×1.0) |
| FedProx | **91.39**(82.32) | 8.97 ↑ | **76**(None) | ×**3.4**(None) | **87.01**(50.01) | 37.00↑ | **127**(831) | ×**7.6**(×1.2) |
| SCAFFOLD | **92.34**(85.31) | 7.03 ↑ | **15**(66) | ×**17.0**(×4.0) | **79.60**(52.76) | 26.84↑ | **171**(627) | ×**5.7**(×1.5) |
| FedNova | **92.85**(86.21) | 6.64 ↑ | **31**(120) | ×**8.4**(×2.2) | **86.64**(45.97) | 40.67↑ | **199**(None) | ×**4.9**(None) |

"Round": communication rounds arriving at the target accuracy. "None": not attaining the target accuracy. Results in "(·)" indicates training without FedFed.

Table 2: Top-1 accuracy with(without) FedFed under different heterogeneity degree, local epochs, and clients number on FMNIST.

| | centralized training ACC = 95.64% w/(w/o) **FedFed** | | | | | | | |
| --- | --- | --- | --- | --- | --- | --- | --- | --- |
| | ACC↑ | Gain↑ | Round↓ | Speedup↑ | ACC↑ | Gain↑ | Round↓ | Speedup↑ |
| | $\alpha = 0.1, E = 1, K = 10$ (Target ACC =86%) | | | | $\alpha = 0.05, E = 1, K = 10$ (Target ACC =78%) | | | |
| FedAvg | **92.34**(86.73) | 5.61↑ | **14**(121) | ×**8.6**(×1.0) | **90.69**(78.34) | 12.35↑ | **16**(420) | ×**26.3**(×1.0) |
| FedProx | **92.09**(87.73) | 4.36 ↑ | **32**(129) | ×**2.1**(×0.9) | **89.68**(82.03) | 7.65↑ | **16**(44) | ×**26.3**(9.5) |
| SCAFFOLD | **91.62**(86.31) | 3.89↑ | **29**(147) | ×**4.2**(× 0.8) | **80.48**(76.63) | 3.85↑ | **139**(None) | ×**6.2**(None) |
| FedNova | **92.39**(87.03) | 5.36↑ | **18**(88) | ×**6.7**(×1.4) | **89.72**(79.98) | 9.74↑ | **16**(531) | ×**26.3**(× 0.8) |
| | $\alpha = 0.1, E = 5, K = 10$ (Target ACC =87%) | | | | $\alpha = 0.1, E = 1, K = 100$ (Target ACC =90%) | | | |
| FedAvg | **92.26**(87.43) | 4.83↑ | **19**(276) | ×**14.5**(×1.0) | **92.71**(90.21) | 2.5↑ | **243**(687) | ×**2.8**(×1.0) |
| FedProx | **91.79**(86.63) | 5.16↑ | **34**(None) | ×**8.1**(None) | **92.82**(90.17) | 2.65↑ | **284**(501) | ×**2.4**(×1.4) |
| SCAFFOLD | **92.92**(87.21) | 5.71↑ | **8**(112) | ×**34.5**(×2.5) | **90.28**(84.87) | 5.41↑ | **952**(None) | ×**0.7** (None) |
| FedNova | **92.30**(87.67) | 4.63↑ | **8**(187) | ×**34.5**(×1.5) | **91.04**(85.32) | 5.72↑ | **589**(None) | ×**1.2**(None) |

*where,* $\hat{\epsilon}_{\hat{\delta}} = \min\{\frac{k\rho\sqrt{R\log(1/\delta)}}{\sigma_s}, \frac{\rho\sqrt{R\log(1/\delta)}}{\sigma_s}[\frac{e^{\rho\sqrt{R\log(1/\delta)}/\sigma_s}-1}{e^{\rho\sqrt{R\log(1/\delta)}/\sigma_s}+1}k + \epsilon(2k\log(e + (\rho\sqrt{kR\hat{\delta}\log(1/\delta)})/(\sigma_s^2\hat{\delta})))^{1/2}], \frac{\rho\sqrt{R\log(1/\delta)}}{\sigma_s} \cdot [\frac{e^{\rho\sqrt{R\log(1/\delta)}/\sigma_s}-1}{e^{\rho\sqrt{R\log(1/\delta)}/\sigma_s}+1}k + \sqrt{2k\log(1/\hat{\delta})}]\}.$

Atop composition theorem [28], Theoreom 3.4 shows that FedFed is able to provide a strong privacy guarantee for FL systems.

## 4 Experiments

We organize this section in the following aspects: a) the main experimental settings that we primarily followed (Sec 4.1); b) the main results and observations of applying FedFed to existing popular FL algorithms (Sec 4.2); c) the sensitivity study of hyper-parameters on effecting model performance (Sec 4.3); and d) empirical analysis of privacy with two kinds of attacks (Sec 4.4).

### 4.1 Experimental Setup

**Federated Non-IID Datasets.** Following previous works [10, 29], we conduct experiments over CIFAR-10, CIFAR100 [30], Fashion-MNIST(FMNIST) [31], and SVHN [32]. Following [5], we employ latent Dirichlet sampling (LDA) [33] to simulate Non-IID distribution.In our experiments, following [5, 34], we set $\alpha = 0.1$ and $\alpha = 0.05$ by LDA. Besides, we evaluate FedFed with other two widely adopted partition strategies: $\#C = k$ [4, 5] and Subset method [9], mainly including label skew and quantity skew (Appendix F.1 shows more detail of data distribution).

Table 3: Top-1 accuracy with(without) FedFed under different heterogeneity degree, local epochs, and clients number on CIFAR-100.

| | centralized training ACC = 75.56% w/(w/o) **FedFed** | | | | | | | |
|---|---|---|---|---|---|---|---|---|
| | ACC↑ | Gain↑ | Round↓ | Speedup↑ | ACC↑ | Gain↑ | Round↓ | Speedup↑ |
| | $\alpha = 0.1, E = 1, K = 10$ (Target ACC =67%) | | | | $\alpha = 0.05, E = 1, K = 10$ (Target ACC =61%) | | | |
| FedAvg | **69.64**(67.84) | 1.8↑ | **283**(495) | ×**1.7** (×1.0) | **68.49**(62.01) | 6.48↑ | **137**(503) | ×**3.7**(×1.0) |
| FedProx | **70.02**(65.34) | 4.68 ↑ | **233**(None) | ×**2.1**(None) | **69.03**(61.29) | 7.74↑ | **141**(485) | ×**3.6**(1.0) |
| SCAFFOLD | **70.14**(67.23) | 2.91↑ | **198**(769) | ×**2.5**(× 0.6) | **69.32**(58.78) | 10.54↑ | **81**(None) | ×**6.2**(None) |
| FedNova | **70.48**(67.98) | 2.5↑ | **147**(432) | ×**3.4**(×1.1) | **68.92**(60.53) | 8.39↑ | **87**(None) | ×**5.8**(None) |
| | $\alpha = 0.1, E = 5, K = 10$ (Target ACC =69%) | | | | $\alpha = 0.1, E = 1, K = 100$ (Target ACC =48%) | | | |
| FedAvg | **70.96**(69.34) | 1.62↑ | **79**(276) | ×**3.5**(×1.0) | **60.58**(48.21) | 12.37↑ | **448**(967) | ×**2.2**(×1.0) |
| FedProx | **69.66**(62.32) | 7.34↑ | **285**(None) | ×**1.0**(None) | **67.69**(48.78) | 18.91↑ | **200**(932) | ×**4.8**(×1.0) |
| SCAFFOLD | **70.76**(70.23) | 0.53↑ | **108**(174) | ×**2.6**(×1.6) | **66.67**(51.03) | 15.64↑ | **181**(832) | ×**5.3**(×1.2) |
| FedNova | **69.98**(69.78) | 0.2↑ | **89**(290) | ×**3.1**(×1.0) | **67.62**(48.03) | 19.59↑ | **198**(976) | ×**4.9**(×1.0) |

Table 4: Top-1 accuracy with(without) FedFed under different heterogeneity degree, local epochs, and clients number on SVHN.

| | centralized training ACC = 96.56% w/(w/o) **FedFed** | | | | | | | |
|---|---|---|---|---|---|---|---|---|
| | ACC↑ | Gain↑ | Round↓ | Speedup↑ | ACC↑ | Gain↑ | Round↓ | Speedup↑ |
| | $\alpha = 0.1, E = 1, K = 10$ (Target ACC =88%) | | | | $\alpha = 0.05, E = 1, K = 10$ (Target ACC =82%) | | | |
| FedAvg | **93.21**(88.34) | 4.87↑ | **105**(264) | ×**2.5**(×1.0) | **93.49**(82.76) | 10.73↑ | **194**(365) | ×**1.9**(×1.0) |
| FedProx | **91.80**(86.23) | 5.574↑ | **233**(None) | ×**1.1**(None) | **93.21**(79.43) | 13.78↑ | **37**(None) | ×**9.9**(None) |
| SCAFFOLD | **88.41**(80.12) | 8.29↑ | **357**(None) | ×**0.**(None) | **90.27**(75.87) | 14.4↑ | **64**(None) | ×**5.7**(None) |
| FedNova | **92.98**(89.23) | 3.75↑ | **113**(276) | ×**2.3**(×1.0) | **93.05**(82.32) | 10.73↑ | **37**(731) | ×**9.9**(×0.5) |
| | $\alpha = 0.1, E = 5, K = 10$ (Target ACC =87%) | | | | $\alpha = 0.1, E = 1, K = 100$ (Target ACC =89%) | | | |
| FedAvg | **93.77**(87.24) | 6.53↑ | **105**(128) | ×**1.2**(×1.0) | **91.04**(89.32) | 1.72↑ | **763**(623) | ×0.8(×**1.0**) |
| FedProx | **91.15**(77.21) | 13.94↑ | **142**(None) | ×**0.9**(None) | **91.41**(88.76) | 2.65↑ | **733**(645) | ×0.8(×**1.0**) |
| SCAFFOLD | **93.78**(80.98) | 12.8↑ | **20**(None) | ×**6.4**(None) | **92.73**(88.32) | 4.41↑ | **507**(687) | ×**1.2**(×0.9) |
| FedNova | **93.66**(89.03) | 4.63↑ | **52**(177) | ×**2.5**(×0.7) | **84.05**(81.87) | 2.18↑ | None(None) | None(None) |

**Models, Metrics and Baselines.** We use ResNet-18 [35] both in the feature distillation and classifier in FL. We evaluate the model performance via two popular metrics in FL: a) communication rounds and b) best accuracy. Typically, the target accuracy is set to be the best accuracy of vanilla FedAvg. As a plug-in approach, we apply FedFed to prevailing existing FL algorithms, such as FedAvg [4], FedProx [6], SCAFFOLD [7], and FedNova [22] to compare the efficiency of our method. More experimental settings and details are listed in Appendix B.1.

## 4.2 Main Results

**Generative Model Selection.** We consider two generative models in this paper to distill raw data, i.e., Resnet Generator, a kind of vanilla generator used in many works [36, 37], and $\beta$-VAE [38], an encoder-decoder structure by variational inference. We verify the effectiveness of FedFed with different generative models. According to results shown in Figure 2 (a), we observe that $\beta$-VAE gets better performance in FedFed. Thus, we mainly report the results of $\beta$-VAE in the rest.

**Result Analysis.** The experimental results on CIFAR-10, CIFAR-100, FMNIST, and SVHN are shown respectively in Tables 1, 2, 3, and 4. It can be seen that the proposed method can consistently and significantly improve model accuracy under various settings. Moreover, FedFed can promote the convergence rate of different algorithms[4]. Specifically, FedFed brings significant performance gain in CIFAR-10 under the $K = 100$ setting shown in Table 1 and also notably speeds up the convergence rate in FMNIST under the $\alpha = 0.05, E = 1, K = 10$ setting. However, attributable to the original FL methods almost reaching a performance bottleneck, FedFed achieves limited performance gain in SVHN and FMNIST. FedFed also has limited improvement in CIFAR-100. A possible reason

---

[4]More supporting figures of convergence rate are moved to appendix F.4

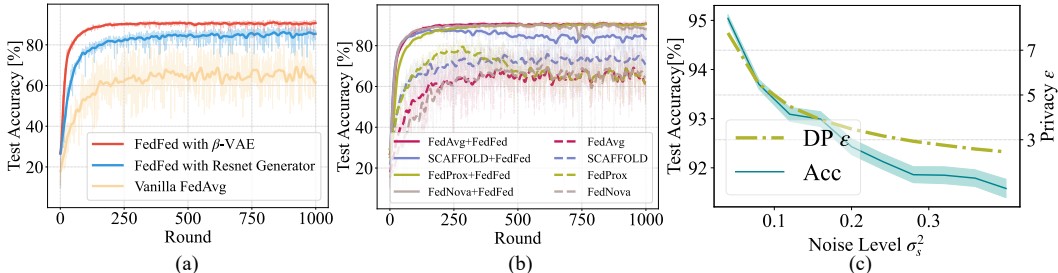

Figure 2: More facts of FedFed. (a) Convergence rate of different generative models (i.e., $\beta$-VAE[38] and ResNet generator[37]) compared with vanilla FedAvg. (b) Test accuracy and convergence rate on different federated learning algorithms with or without FedFed under $\alpha = 0.1, E = 1, K = 100$. (c) Test accuracy on FMNIST with different noise level $\sigma_s^2$ in Theorem 3.3, obtaining various privacy $\epsilon$ (lower $\epsilon$ is preferred). As the noise increased, the level of protection gradually increased.

Table 5: Experiment results of different Non-IID partition methods on CIFAR-10 with 10 clients.

| Partition Method | Test Accuracy w/(w/o) **FedFed** | | | |
|---|---|---|---|---|
| | FedAvg | FedProx | SCAFFOLD | FedNova |
| $\alpha = 0.1$ | **92.34**(79.35) | **92.12**(83.06) | **89.66**(83.67) | **92.23**(80.95) |
| #$C = 2$ | **89.23**/42.54 | **88.17**/58.45 | **84.43**/46.82 | **89.54**/45.42 |
| Subset | **90.29**/39.53 | **89.11**/32.87 | **89.92**/35.26 | **90.00**/38.52 |

Table 6: Experiment results with different noise adding on CIFAR-10.

| Noise Type | Test Accuracy on Different Noise with **FedFed** | | | |
|---|---|---|---|---|
| | FedAvg | FedProx | SCAFFOLD | FedNova |
| Gaussian Noise | 92.34 | 92.12 | 89.66 | 92.23 |
| Laplacian Noise | 92.30 | 91.36 | 91.24 | 91.73 |

is that existing methods can achieve performance comparable to centralized training. Moreover, we conducted an additional experiment involving sharing the full data with DP protection. The results, presented in Appendix F.3, indicate that sharing the full data with DP protection leads to a degradation in the performance of the FL system. This degradation occurs because protecting the full data necessitates a relatively large noise to achieve the corresponding protection strength required to safeguard performance-sensitive features.

**Surprising Observations.** We find that training among 100 clients in CIFAR-10 and CIFAR100 reaches a significant improvement (e.g., at most $40.67\%$!). A possible reason is that the missed data knowledge can be well replenished by FedFed. Moreover, all methods paired with FedFed under various settings can achieve similar prediction accuracies, demonstrating that FedFed endows FL models robustness against data heterogeneity. Table 5 shows that two kinds of heterogeneity partition cause more performance decline than LDA ($\alpha = 0.1$). Yet, FedFed attains noteworthy improvement, indicating the robustness against Non-IID partition.

### 4.3 Ablation Study

**DP Noise.** To explore the relationship between privacy level $\epsilon$ and prediction accuracy, we conduct experiments with different noise levels $\sigma_s^2$. As shown in Figure 2(c), the prediction accuracy decreases with increasing noise level (more results in Appendix F.5). To verify whether the FedFed is robust against the selection of noise, we also consider Laplacian noise in applying DP to privacy protection. The results of Laplacian noise are reported in Table 6, demonstrating the robustness of FedFed.

**Hyper-parameters.** We further evaluate the robustness of FedFed against hyper-parameters. During the feature distillation process, as the constraint parameter $\rho$ in Eq. 9 decreases, the DP strength to protect performance-sensitive features decreases, and the performance of the global model decreases, owing to the less information contained in performance-sensitive features.

### 4.4 Privacy Verification

Besides the theoretical analysis, we provide empirical analysis to support the privacy guarantee of FedFed. We wonder whether the globally shared data can be inferred by some attacking methods. Thus, we resort to model inversion attack [39], widely used in the literature to reconstruct data. The results[5] in Figure 3 (a) and (b) indicate that FedFed could protect globally shared data. We also conduct another model inversion attack [40] and the results can be found in Appendix E.1.

---

[5]More details and results can be found in Appendix E.1.

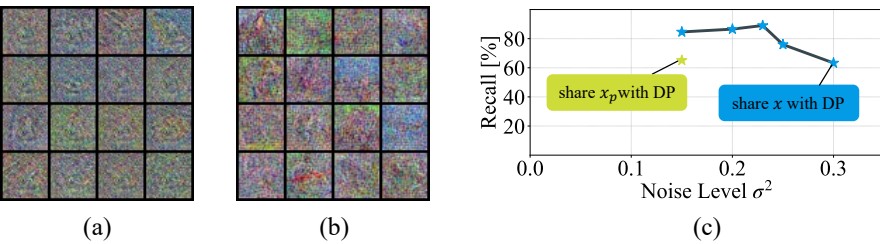

|  (a)  |  (b)  |  (c)  |

Figure 3: Attack results on FedFed. (a) shows the protected data $\mathbf{x}_p$. (b) reports the model inversion attacked data. (c) shows results of membership inference attacks: the green star represents the recall for FedFed, while the blue stars show the searching process varying with $\sigma_s^2$ for sharing raw data.

Additionally, we perform membership inference attack [41] to illustrate the difference between FedFed and sharing all data features with DP protection. The results illustrated in Figure 3 (c) show that noise level $\sigma^2 = 0.3$ over raw data $\mathbf{x}$ can achieve similar protection as noise $\sigma_s^2 = 0.15$ over globally shared data $\mathbf{x}_p$, which aligns with Theorem 3.3. More details can be found in Appendix E.2

## 5  Related Work

Federated Learning (FL) models typically perform poorly when meeting severe Non-IID data [2, 7]. To tackle data heterogeneity, advanced works introduce various learning objectives to calibrate the updated direction of local training from being far away from the global model, e.g., FedProx[6], FedIR [42], SCAFFOLD [7], and MOON [43]. Designing model aggregation schemes like FedAvgM [33], FedNova [22], FedMA [44], and FedBN [45] shows the efficacy on heterogeneity mitigation. Another promising direction is the information-sharing strategy, which mainly focuses on synthesizing and sharing clients' information to mitigate heterogeneity [9, 46, 47]. To avoid exposing privacy caused by the shared data, some methods utilize the statistics of data [48], representations of data [12], logits [49, 10], embedding [50]. However, advanced attacks pose potential threats to methods with data-sharing strategies [51].

FedFed is inspired by [34], where pure noise is shared across clients to tackle data heterogeneity. In this work, we relax the privacy concern by sharing partial features in the data with DP protection. In addition, our work is technically similar to an adversarial learning approach [52]. Our method distinguishes by defining various types of features and delving into the exploration of data heterogeneity within FL. More discussion can be found in Appendix G.

## 6  Conclusion

In this work, we propose a novel framework called FedFed to tackle data heterogeneity in FL by employing the promising information-sharing approach. Our work extends the research line of constructing shareable information of clients by proposing to share partial features in data. This shares a new route of improving training performance and maintaining privacy. Furthermore, FedFed has served as a source of inspiration for a new direction focused on improving performance in open-set scenarios [53]. Another avenue of exploration involves deploying FedFed in other real-time FL application scenarios, such as recommendation systems [54] and healthcare system [55], to uncover its potential benefits.

**Limitation.** In reality, limited local hardware resources may limit the power of FedFed, since FedFed introduces some extra overheads like communication and storage overheads. We leave it as future works to explore a hardware-friendly version or real-world application [56, 57]. Additionally, FedFed raises potential privacy concerns that we leave as a future research exploration, such as integrating cryptography [58, 59]. We anticipate that our work will inspire further investigations to comprehensively evaluate the privacy risks associated with information-sharing strategies aimed at mitigating data heterogeneity.

## Acknowledgments and Disclosure of Funding

We thank the reviewers for their valuable comments. Hao Peng was supported by the National Key R&D Program of China through grant 2021YFB1714800, NSFC through grants 62002007, U21B2027, 61972186, and 62266027, S&T Program of Hebei through grant 20310101D, Natural Science Foundation of Beijing Municipality through grant 4222030, and the Fundamental Research Funds for the Central Universities, Yunnan Provincial Major Science and Technology Special Plan Projects through grants 202103AA080015, 202202AD080003 and 202303AP140008, General Projects of Basic Research in Yunnan Province through grant 202301AS070047. Yonggang Zhang and Bo Han were supported by the NSFC Young Scientists Fund No. 62006202, NSFC General Program No. 62376235, Guangdong Basic and Applied Basic Research Foundation No. 2022A1515011652, CCF-Baidu Open Fund, HKBU Faculty Niche Research Areas No. RC-FNRA-IG/22-23/SCI/04, and HKBU CSD Departmental Incentive Scheme. Tongliang Liu was partially supported by the following Australian Research Council projects: FT220100318, DP220102121, LP220100527, LP220200949, and IC190100031. Xinmei Tian was supported in part by NSFC No. 62222117.

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

# A    Overview of FedFed Framework

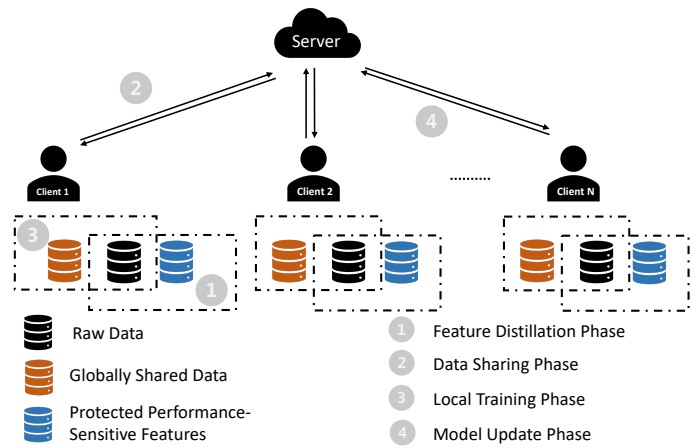

Figure 4: Overview of FedFed

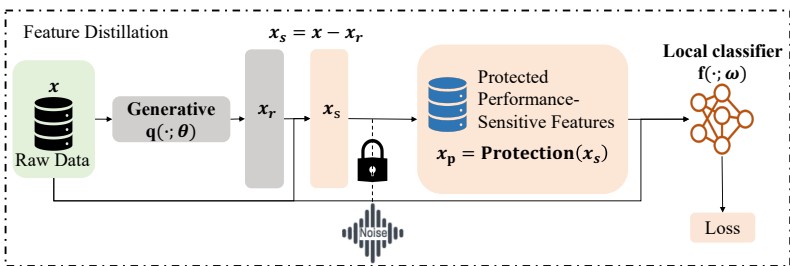

Figure 5: The Pipeline of Feature Distillation

The training process of the federated learning system with FedFed can be summarized into four phases. Firstly, every participant of FL distils the local private raw data $\mathbf{x}$ and decouples $\mathbf{x}$ into performance-robust features $\mathbf{x}_r$ and performance-sensitive features $\mathbf{x}_s$ during the feature distillation phase. Then all clients send $\mathbf{x}_p$ to construct a global dataset, $\mathbf{x}_s$ with DP protection and keep $\mathbf{x}_r$ locally. In the local training phase, the selected clients sample a subset from the globally shared dataset and update the local model with local raw data $\mathbf{x}$ and sampled information $\mathbf{x}_p$ from server. Finally, clients upload the latest local model and aggregate a global model based on a certain aggregation algorithm, like the weighted average strategy in FedAvg. Figure 4 shows the overview of FedFed framework, while Figure 5 depicts its pipeline of feature distillation. The loss function of feature distillation in Figure 5 can be formulated as Eq (9).

# B    Federated Learning Algorithms with FedFed

## B.1    Implementation Details

We list all relevant parameters in this paper in Table 7. We fine-tune learning rate in the set of $\{0.0001, 0.001, 0.01, 0.1\}$ and report the best results and corresponding learning rate. Whereas, we use 0.01 as the learning rate. In our work, we deploy FedFed on SCAFFOLD and set the learning rate $\eta_k$ to 0.0001 for the SVHN dataset. Furthermore, we use $\eta_k = 0.001$ in SVHN while FedNove is deployed with our method.

The batch size is set as 64 when $K = 10$ and 32 for $K = 100$. On the server side, we select 5 clients for aggregation per round when $K = 10$, and 10 clients per round when $K = 100$. This corresponds to a sampling rate of 50% for 10 clients ($K = 10$) and 10% for 100 clients ($K = 100$). The noise level in our experiments is $\mathcal{N}(0, 0.15)$. Besides, all experiments are performed on Python 3.8, 36 core 3.00GHz Intel Core i9 CPU, and NVIDIA RTX A6000 GPUs.

Table 7: The values of all parameters in this paper

| Symbolic representation | Value | Description |
|---|---|---|
| **Federated Learning Relevant** | | |
| $\alpha$ | 0.1/0.05 | heterogeneity degree |
| $T_d$ | 15 | communication round of feature distillation |
| $T_r$ | 1,000 | communication round of classifier training |
| $E_d$ | 1 | local epochs of feature distillation |
| $E/E_r$ | 1/5 | local epochs of classifier training |
| $\sigma_s^2$ | 0.15 | DP noise level, added to $\mathbf{x}_s$ |
| $|C_t|/|C_r|$ | 5/10 | #selected clients every communication round |
| K | 10/100 | #clients of federated system |
| **Training Process Relevant** | | |
| $\eta_k$ | 0.01/0.001/0.0001 | learning rate |
| $B$ | 32/64 | batch size |
| $M$ | 0.9 | momentum |
| wd | 0.0001 | weight decay for regularization |

"#" represents the number of, and "/" denotes we try these values for these symbolic representations.

### B.2 Pseudo-code and Explaination

In this section, we give the pseudo-code of FL algorithm with FedFed. Algorithm 2 gives the procedure of FedAvg and FedProx with FedFed. The difference between FedAvg and FedProx is the local objective function, and we highlight it in the following equation:

$$\min_{\phi_k} \mathbb{E}_{(\mathbf{x},\mathbf{y}) \sim P(X_k, X_k)} \ell(f(\mathbf{x}; \phi_k), \mathbf{y}) + \mathbb{E}_{(\mathbf{x}_p, \mathbf{y}) \sim P(\mathbf{h}(X_k), Y_k)} \ell(f(\mathbf{x}_p; \phi_k), \mathbf{y}) + \frac{\mu}{2} ||\phi - \phi^r||^2, \quad (11)$$

SCAFFOLD attempts to estimate the client shift degree to correct client update direction. Particularly, it works through server control variate $c$ and client control variate $c_k$ of client k. SCAFFOLD with FedFed is summarized in Algorithm 3. And FedNova is proposed to resolve the objective inconsistency for defying data heterogeneity. The outline of our plugin deploys on FedNova in the Algorithm 4. In accordance with our process of how to apply FedFed, every participant of the FL system can deploy our framework easily without any modification of objective function or aggregation scheme in their own methods.

## C Full Analysis on Information Theory Perspective

### C.1 Features Partition

From an information theory perspective, we present formal definitions, namely Definition 3.1 and Definition 3.2, to define performance-robust features and performance-sensitive features. Specifically, Definition 3.1 captures the desirable attributes when we partition the features, which are abstracted to be a variable in general. A valid partition should maintain all information of the original variable $X$ losslessly. That is, neither extra information is introduced nor key information is lost. So we can get (i) and (ii) in Definition 3.1. After determining the measure space, we partition the $X$ to be $X = X_1 + X_2$. The mutual information of $X_1$ and $X_2$ is none. Here, $X_1$ and $X_2$ can be symmetric. Or saying, there is no overlap between $X_1$ and $X_2$, as described by (iii) in Definition 3.1.

---

**Algorithm 2** FedAvg/FedProx with FedFed

---

**Server Input:** initial global model $\phi^0$, communication round $T_r$.
**Client $k$'s Input:** local epochs $E_r$, local private datasets $\mathcal{D}^k$, learning rate $\eta_k$.
  **Initialization:** server distributes the initial model $\phi^0$ to all clients,
  Generate globally shared dataset $\mathcal{D}^s$. $\leftarrow$ Detail in Algorithm 1

  Distribute $\mathcal{D}^s$ to all clients and $\mathcal{D}_t^k = \mathcal{D}^k \cup \mathcal{D}^s$.
  **Server Executes:**
  **for** each round $r = 1, 2, \cdots, T_r$ **do**
    server samples a subset of clients $C_r \subseteq \{1, ..., K\}$
    server **communicates** $\phi^r$ to selected clients $k \in C_r$
    **for** each client $k \in C_r$ **in parallel do**
      $\phi_k^{r+1} \leftarrow$ Client_Training($k, \phi^r$)
    **end for**
    $\phi^{r+1} \leftarrow AGG(\phi_k^{r+1})$
  **end for**

  **Client_Training($k, \phi^r$):**
  $\phi^r$ initialize local model $\phi_k^r$
  **for** each local epoch $e$ with $e = 1, 2, \cdots, E_r$ **do**
    $\phi_k^{r+1} \leftarrow$ SGD update with $\mathcal{D}_t^k$ using Eq. (10), FedProx uses Eq. (11)
  **end for**
  **Return** $\phi_k^{r+1}$ to server

---

Built upon the Definition 3.1, we present the definition of performance-sensitive features and performance-robust features, as in Definition 3.2. Intuitively, performance-sensitive features contain all label information, while performance-robust features contains all information about the data except for the label information. That is, the features to be partitioned are either performance-sensitive features or performance-robust features.

### C.2 FedFed on Information Bottleneck Perspective

In this section, we analyse FedFed under an information bottleneck(IB) aspect. In conventional, information bottleneck [21] can be formalized as:

$$\min_Z I(X; Y|Z), \ s.t. \ I(X; Z) \leq I_c. \tag{12}$$

where to restrict the complexity of encoding $Z$. Information entropy is denoted as $H(\cdot)$ and $I(\cdot; \cdot)$ is the mutual information of two variables. In FedFed, our competitive mechanism stems from IB getting the following definition:

$$
\begin{aligned}
&\min_Z I(X; Y|Z), \ s.t. \ I(X - Z; X|Z) \geq I_c, \\
\Leftrightarrow \ &\min_Z H(Y|Z), \ s.t. \ H(Z) \leq H_c, \\
\Leftrightarrow \ &\min_{\mathbf{z}} -\mathbb{E} \ \log p(\mathbf{y}|\mathbf{z}), \ s.t. \ ||\mathbf{z}||_2^2 \leq \rho.
\end{aligned}
\tag{13}
$$

According to our framework, $\mathbf{z}$ can be defined as: $\mathbf{z} = \mathbf{z} - \mathbf{x}_r = \mathbf{x} - q(\mathbf{x}; \theta)$, and then we can the get the optimization view of FedFed:

$$\min_\theta -\mathbb{E}_{(\mathbf{x}, \mathbf{y}) \sim P(X_k, Y_k)} \log p(\mathbf{y}|\mathbf{x} - q(\mathbf{x}; \theta)), \ s.t. \ ||\mathbf{x} - q(\mathbf{x}; \theta)||_2^2 \leq \rho. \tag{14}$$

Ultimately, in an information bottleneck perspective, $\min_\theta -\mathbb{E}_{(\mathbf{x}, \mathbf{y}) \sim P(X_k, Y_k)} \log p(\mathbf{y}|\mathbf{x} - q(\mathbf{x}; \theta))$ intend to contain more information that can be used for generalization. $||\mathbf{x} - q(\mathbf{x}; \theta)||_2^2 \leq \rho$ aims at distilling raw data for minimum features which means maximal compression of $\mathbf{x}_s$.

**Algorithm 3** SCAFFOLD with FedFed

---

**Server Input:** initial global model $\phi^0$, communication round $T_r$, server control variate $c = 0$.
**Client $k$'s Input:** local epochs $E_r$, local private datasets $\mathcal{D}^k$, learning rate $\eta_k$, variate $c_k = 0$.

   **Initialization:** server distributes the initial model $\phi^0$ to all clients,

   Generate globally shared dataset $\mathcal{D}^s$. $\leftarrow$ Detail in Algorithm 1

   Distribute $\mathcal{D}^s$ to all clients and $\mathcal{D}_t^k = \mathcal{D}^k \cup \mathcal{D}^s$.

   **Server Executes:**
   **for** each round $r = 1, 2, \cdots, T_r$ **do**
      server samples a subset of clients $C_r \subseteq \{1, ..., K\}$
      server **communicates** $\phi^r$ to selected clients $k \in C_r$
      **for** each client $k \in C_r$ **in parallel do**
         $\Delta\phi_k^{r+1}, \Delta c \leftarrow$ Client_Training$(k, \phi^r, c)$
      **end for**
      $\phi^{r+1} \leftarrow AGG(\Delta\phi_k^{r+1}), \quad c \leftarrow c + \frac{|C_r|}{K}\Delta c$
   **end for**

   **Client_Training**$(k, \phi^r, c)$**:**
   $\phi^r$ initialize local model $\phi_k^r$
   **for** each local epoch $e$ with $e = 1, 2, \cdots, E_r$ **do**
      $\phi_k^{r+1} \leftarrow \phi_k^r - \eta_k[\nabla_{\phi_k^r} L_k(\phi_k^r) - c_k + c]$, (SGD update with $\mathcal{D}_t^k$ use Eq. (10))
   **end for**
   $c_k^\star \leftarrow (i)\nabla_{\phi_k^r} L_k(\phi_k^r), or (ii) c_k - c + \frac{1}{E\eta_k}(\phi_k^r - \phi_k^{r+1})$
   $\Delta c \leftarrow c_k^\star - c_k, \quad \Delta\phi_k^{r+1} \leftarrow \phi_k^{r+1} - \phi^r$
   $c_k \leftarrow c_k^\star$
   **Return** $(\Delta\phi_k^{r+1}, \Delta c)$ to server

---

*Proof.* To get Eq (13), we firstly have:

$$
\begin{aligned}
I(X;Y|Z) &= I(X;Y) - I(X;Y;Z)\\
&= H(X) + H(Y) - H(X,Y) - [I(Y;Z) - I(Y;Z|X)]\\
&= H(X) + H(Y) - H(X,Y) - [H(Y) + H(Z) - H(Y,Z)]\\
&\quad + [H(Y|X) - H(Y|X,Z)]\\
&= H(X) + H(Y) - H(X,Y) - H(Y) - H(Z)\\
&\quad + H(Y,Z) + H(Y|X) - H(Y|X,Z)\\
&= H(X) - H(X,Y) - H(Z) + H(Y,Z) + H(Y|X) - H(Y|X,Z)\\
&= H(Y,Z) - H(Z) - H(Y|X,Z)\\
&= H(Z) + H(Y|Z) - H(Z) - H(Y|X,Z)\\
&= H(Y|Z) - H(Y|X,Z)\\
&= H(Y|Z) - H(Y|X).
\end{aligned}
$$

$$
\begin{aligned}
I(X - Z; X|Z) &= I(X - Z; X) - I(X - Z; X; Z)\\
&= H(X - Z) + H(x) - H(X - Z, X) - [I(X;Z) - I(X;Z|X - Z)]\\
&= H(X - Z) + H(X) - H(X - Z, X) - [H(X) + H(Z) - H(X,Z)]\\
&\quad + [H(X|X - Z) - H(X|X - Z, Z)]\\
&= H(X - Z) + H(X) - H(X - Z, X) - H(X) - H(Z)\\
&\quad + H(X,Z) + H(X|X - Z) - H(X|X - Z, Z)\\
&= H(X - Z) - H(X - Z, X) - H(Z) + H(X,Z) + H(X|X - Z)\\
&= H(X - Z) - H(X) - H(Z) + H(X) + H(X|X - Z)\\
&= H(X - Z) - H(Z) + H(X|X - Z)\\
&= H(X, X - Z) - H(Z)\\
&= H(X) - H(Z).
\end{aligned}
$$

$$I(X - Z; X|Z) \geq I_c \Leftrightarrow H(X) - H(Z) \geq I_c \Rightarrow H(Z) \leq H(x) - I_c = H_c \Rightarrow H(Z) \leq H_c.$$

$$\tag{15}$$

**Algorithm 4** FedNova with FedFed

---

**Server Input:** initial $\phi^0$, communication round $T_r$.
**Client $k$'s Input:** Epochs $E_r$, local private datasets $\mathcal{D}^k$, learning rate $\eta_k$, momentum factor $\varrho_k$.

  **Initialization:** server distributes the initial model $\phi^0$ to all clients,

  Generate globally shared dataset $\mathcal{D}^s$. $\leftarrow$ Detail in Algorithm 1

  Distribute $\mathcal{D}^s$ to all clients and $\mathcal{D}_t^k = \mathcal{D}^k \cup \mathcal{D}^s$

  **Server Executes:**
  **for** each round $r = 1, 2, \cdots, T_r$ **do**
    server samples a subset of clients $C_r \subseteq \{1, ..., K\}$
    server **communicates** $\phi^r$ to selected clients $k \in C_r$
    **for** each client $k \in C_r$ **in parallel do**
      $\Delta\phi_k^{r+1}, a_k \leftarrow$ Client_Training$(k, \phi^r)$
    **end for**
    $\phi^{r+1} \leftarrow \phi^r - \frac{\sum_{k \in C_r} a_k}{K} \sum_{k \in C_r} \frac{\Delta\phi_k^{r+1}}{K}$
  **end for**

  **Client_Training$(k, \phi^r)$:**
  $\phi^r$ initialize local model $\phi_k^r$
  **for** each local epoch $e$ with $e = 1, 2, \cdots, E_r$ **do**
    $\phi_k^{r+1} \leftarrow$ SGD updata with $\mathcal{D}_t^k$ use Eq. (10))
  **end for**
  $a_k \leftarrow [E - \varrho_k(1 - \varrho^E)/(1 - \varrho_k)]/(1 - \varrho_k)$
  $\Delta\phi_k^{r+1} \leftarrow (\phi_k^{r+1} - \phi_k^r)/(\eta_k a_k)$
  **Return** $\Delta\phi_k^{r+1}, a_k$ to server

---

Then, we get $\min_Z I(X; Y|Z)$, *s.t.* $I(X - Z; X|Z) \geq I_c \Leftrightarrow \min_Z H(Y|Z)$, *s.t.* $H(Z) \leq H_c$.

Furthermore, we have:

$$H(Y|Z) = \int_{\mathbf{z}} p(z)H(Y|Z = \mathbf{z})d\mathbf{z} = \int_{\mathbf{z}} p(\mathbf{z}) \int_{\mathbf{y}} p(\mathbf{y}|\mathbf{z}) \log \frac{1}{p(\mathbf{y}|\mathbf{z})} d\mathbf{y}d\mathbf{z}$$

$$= -\int_{\mathbf{z}} \int_{\mathbf{y}} p(\mathbf{y}, \mathbf{z}) \log p(\mathbf{y}|\mathbf{z})d\mathbf{y}d\mathbf{z}$$

$$= -\mathbb{E} \log p(\mathbf{y}|\mathbf{z}).$$

$$
\begin{align}
H(Z) &\leq \frac{1}{2} \sum_{i=1}^d \ln Var(\mathbf{z}_i)2\pi e = \frac{1}{2} \sum_{i=1}^d \ln Var(\mathbf{z}_i) + \frac{1}{2} \sum_{i=1}^d \ln 2\pi e \tag{16}\\
&\leq \frac{1}{2} \sum_{i=1}^d \ln Var(\mathbf{z}_i) + \frac{1}{2}d \ln 2\pi e = \frac{1}{2} \sum_{i=1}^d \ln \mathbb{E}(\mathbf{z}_i^2) + \frac{1}{2}d \ln 2\pi e\\
&= \frac{1}{2} \ln \mathbb{E}(\sum_{i=1}^d \mathbf{z}_i^2) + \frac{1}{2}d \ln 2\pi e \leq H_c\\
&\Rightarrow \mathbb{E}||\mathbf{z}||_2^2 \leq e^{2H_c - d \ln 2\pi e} = \rho,
\end{align}
$$

which completes the proof.

## D   Full Analysis on Differentially Private Features

Our security analyses contain two steps, saying, differential privacy for each client and the overall analyses for FedFed. Built upon previous works [27, 16], we derive Lemma D.1 below. For FedFed, we attain Lemma D.2 by limiting $\sigma_r \to \infty$.

**Lemma D.1.** *For sharing raw features* $\mathbf{x}$, $(\epsilon, \delta)$*-DP holds if* $\epsilon' = O(\sqrt{R \log(1/\delta)}(\rho/\sigma_s + (1 - \rho)/\sigma_r))$.

**Lemma D.2.** *For sharing performance-sensitive features* $\mathbf{x}_s$, $(\epsilon, \delta)$-*DP holds if* $\epsilon = O(\rho\sqrt{R\log(1/\delta)}/\sigma_s)$.

## D.1 Analysis and Proof for Theorem 3.3

Recall that DP-SGD [27] samples i.i.d. noise from unbias Gaussian distribution $\mathcal{N}(\mathbf{0}, S_f^2\sigma^2)$, where $S_f$ is the sensitivity. At the algorithmic level, the noise sampled from $\mathcal{N}(\mathbf{0}, \sigma^2 C^2 \mathbf{I})$ is added to a batch of gradients handled by clipping value $C$.

For aligning analyses with conventional DP guarantees, we adopt similar notations and expressions. Specifically, we denote the noise distribution operated on $\mathbf{x}_r$ to be $\mathcal{N}(\mathbf{0}, \sigma_r^2 \mathbf{I})$ and the noise distribution operated on $\mathbf{x}_s$ to be $\mathcal{N}(\mathbf{0}, \sigma_s^2 \mathbf{I})$. Notably, FedFed only adds DP noise to $\mathbf{x}_s$ in practice. Since $\mathbf{x}_r$ is kept by the corresponding client, an adversary could attain nothing (or be regarded as random data without any entropy). This could be regarded as adding a sufficiently large noise on $\mathbf{x}_r$, thus we consider the $\sigma_r \to \infty$. As for $\mathbf{x}_s$, we employ the $\ell_2$-norm clipping when selecting the noise level $\sigma_s$.

Now, let's come back to the setting of sharing all raw data $\mathbf{x}$. We still regard $\mathbf{x}$ to be two parts, i.e., $\mathbf{x} = \mathbf{x}_r + \mathbf{x}_s$ for the latter comparison to FedFed. This trick is motivated by cryptographic standard proof with ideal/real paradigm [60]. For both $\mathbf{x}_r$ and $\mathbf{x}_s$, the noise addition follows DP-SGD's idea, except for adding noise to the sampled data for updating gradients, rather than the latter-computed real gradients. We remark that this method works for FL but may be different for centralized training (with identical training set in the whole training process).

Using the property of post-processing [16], the noise added to the data passes to (or projects to) the following gradient updates. For the view of each client, we know that Theorem D.3[6] holds for $\sigma$ operated on $\mathbf{x}$ from [27].

**Theorem D.3.** *There exist constants $c_1$ and $c_2$ so that given the sampling probability $q = L/N$ and the number of steps $T$, for any $\epsilon < c_1 q^2 T$. The algorithm of DP-SGD is $(\epsilon, \delta)$-differentially private for any $\delta > 0$ if we choose $\sigma = c_2 \frac{q\sqrt{T\log(1/\delta)}}{\epsilon}$.*

For strong composition theorem, the choice of $\sigma$ in Theorem D.3 is $O(q\sqrt{T\log(1/\delta)}/\epsilon)$. In DP-SGD, the $T$ denotes the number of training steps, which means the times of using/sampling private data for DP training. Such meaning represents communication rounds in FL, in which each client receives globally shared data at each round of communication. In the FedFed, we denote the number of communication rounds to be $R$. Now, let's handle the $q$, where $q$ represents the ratio of lot size and the inputting dataset size. In DP-SGD, the per-lot data is directly sampled from the database of size $N$. In other words, a lot of data is the subset of the input dataset. In the FedFed, the training data fed to each client's model is the same as the database size, i.e., $q = N/N = 1$.

DP-SGD asks for the generality and thus bounds $\|f(\cdot)\| = \mathbf{e}$. Yet, FedFed sets $\sigma \propto \|\mathbf{x}_s\|$, which does not ask for a general case. We use the notation $\|\mathbf{x}_s\|_2$ to represent the domain of performance-sensitive features. On the other hand, $\|\mathbf{x}\|_2$ represents the domain of all features (namely, the "domain of datasets"). We additionally borrow the definition of $l_2$-norm and move $\|\mathbf{x}_s\|$ out in the proof. By Equation 8, the client takes $\rho\|\mathbf{x}\|$ for selecting $\sigma$. Since we aim to make an asymptotic analysis here, we ignore the scalars and constants here. Thus, we attain $O(\rho\sqrt{R\log(1/\delta)}/\epsilon)$ for selecting $\sigma$. By rearranging variables, we have $\epsilon = \rho\sqrt{R\log(1/\delta)}/\sigma$ for a loose analysis.

Before continuing the proof, we derive Lemma D.4 to get the relation between $\|\mathbf{x}_r\|$ and $\|\mathbf{x}\|$ for latter usage.

**Lemma D.4.** *Let $\mathbf{x}_r = \mathbf{x} - \mathbf{x}_s$. Then, it holds that $(1-\rho)\|\mathbf{x}\| \leq \|\mathbf{x}_r\| \leq (1+\rho)\|\mathbf{x}\|$ if $\|\mathbf{x}_s\| = \rho\|\mathbf{x}\|$ satisfies.*

*Proof of Lemma D.4:* $\|\mathbf{x} - \mathbf{x}_r\| = \rho\|\mathbf{x}\| \Rightarrow \|\mathbf{x} - \mathbf{x}_r\|^2 = \rho^2\|\mathbf{x}\|^2 \Rightarrow \|\mathbf{x}\|^2 + \|\mathbf{x}_r\|^2 - 2\mathbf{x}^\top\mathbf{x}_r = \rho^2\|\mathbf{x}\|^2 \Rightarrow \|\mathbf{x}\|^2 + \|\mathbf{x}_r\|^2 - 2\|\mathbf{x}\|\|\mathbf{x}_r\|\cos\langle\mathbf{x},\mathbf{x}_r\rangle = \rho^2\|\mathbf{x}\|^2$. Since $\|\mathbf{x}\| > 0$ and $\|\mathbf{x}_r\| > 0$, we get,

$$\|\mathbf{x}\|/\|\mathbf{x}_r\| + \|\mathbf{x}_r\|/\|\mathbf{x}\| - 2\cos\langle\mathbf{x},\mathbf{x}_r\rangle = \rho^2\|\mathbf{x}\|/\|\mathbf{x}_r\|.$$

Let $\|\mathbf{x}_r\| = \alpha\|\mathbf{x}\|$, and our objective is to get the $\alpha$. By replacing $\|\mathbf{x}\|/\|\mathbf{x}_r\|$ and $\|\mathbf{x}_r\|/\|x\|$ with $1/\alpha$ and $\alpha$ respectively, we get,

$$1/\alpha + \alpha - 2\cos\langle\mathbf{x},\mathbf{x}_r\rangle = \rho^2/\alpha.$$

---

[6]Previously "$\geq$", we take the special case "$=$" the same as DP-SGD.

Since $-1 \le \cos\langle \mathbf{x}, \mathbf{x}_r \rangle \le 1$, we get,

$$-2 \le 1/\alpha + \alpha - \rho^2/\alpha \le 2 \Rightarrow -2\alpha \le 1/ + \alpha^2 - \rho^2 \le 2\alpha.$$

Since $0 < \rho < 1$, we take $1/ + \alpha^2 - \rho^2 \le 2\alpha$ and get $(\alpha - 1)^2 \le \rho^2$. Then, we get the meaningful answers $-\rho \le \alpha - 1 \le \rho \Rightarrow 1 - \rho \le \alpha \le 1 + \rho$. Therefore, we attain,

$$(1 - \rho)\|\mathbf{x}\| \le \alpha\|\mathbf{x}\| \le (1 + \rho)\|\mathbf{x}\| \Rightarrow (1 - \rho)\|\mathbf{x}\| \le \|\mathbf{x}_r\| \le (1 + \rho)\|\mathbf{x}\|.$$

Following above, we can establish the relationship between $\|\mathbf{x}\|_2$ and $\|\mathbf{x}_s\|_2$ as follows:

$$\|\mathbf{x}_s\|_2 \le \rho\|\mathbf{x}\|_2.$$

And the relation between noise level $\sigma_s$ and $\mathbf{x}_s$ is:

$$\sigma_s \propto \|\mathbf{x}_s\|_2 \le \rho\|\mathbf{x}\|_2 = \rho M,$$

where we re-scale data into $[0, 1]$ and assume the $\ell_2$-norm of data is less than $M > 0$. To ensure the $\|\mathbf{x}_e\|_2 \le \rho M$, we clip the norm for each sample, which is widely used in DP [27]. $\qquad \square$

Now, let's go back to derive Lemma D.1 and Lemma D.2, and continue the proof. For sharing raw data $\mathbf{x}$, we employ the same method as aforementioned. Since $\|\mathbf{x}_s\| = \rho\|\mathbf{x}\|$, we get $\epsilon'_e = \rho\sqrt{R\log(1/\delta)}/\sigma_s$. By Lemma D.4, we know that $(1 - \rho) \le \frac{\|\mathbf{x}_r\|}{\|\mathbf{x}\|} \le (1 + \rho)$ holds. Thus, we attain,

$$(1 - \rho)\sqrt{R\log(1/\delta)}/\sigma_r \le \epsilon'_r \le (1 + \rho)\sqrt{R\log(1/\delta)}/\sigma_r.$$

Here, we take $\epsilon'_r \ge (1 - \rho)\sqrt{R\log(1/\delta)}/\sigma_r$ since we expect a loose case. By combining $\epsilon'_e$ and $\epsilon'_r$, we attain,

$$\epsilon' = O(\sqrt{R\log(1/\delta)}(\rho/\sigma_s + (1 - \rho)/\sigma_r))$$

for sharing raw $x$, as Lemma D.1. Now, let's take FedFed into consideration. The major difference is that $\mathbf{x}_r$ is kept locally in the whole training process. The root cause lies in that the $\mathbf{x}_r$ owned by the client is privacy-*insensitive* to this corresponding client. We regard $\sigma_r$ operated on $\mathbf{x}_r$ to be a sufficiently large value. Thus, given $\rho \ll 1$, we get,

$$\lim_{\sigma_r \to \infty} \epsilon_r = (1 - \rho)\sqrt{R\log(1/\delta)}/\sigma_r \to 0.$$

That is, for sharing $\mathbf{x}_s$ in FedFed, we have $\epsilon = \epsilon_s = O(\rho/\sigma_s\sqrt{\log(\frac{1}{\delta})})$, as Lemma D.2.

Let's conversely think Lemma D.1 and Lemma D.2. Now, the proof in the following is very straightforward. If we take identical $\sigma_s$ operated on $\mathbf{x}_s$ for FedFed and sharing raw $x$. We know that $\epsilon' > \epsilon$. Conversely, if we take identical $\epsilon, \epsilon'$, we should increase $1/\sigma_s$ and thus reduce $\sigma_s$, given constant $\rho\sqrt{R\log(1/\delta)}$. Thus, for protecting training data in FL, we can attain Theorem 3.3 to summarize the superiority of FedFed when asking for an identical privacy guarantee. Theorem 3.3 explains the reason for the superior model performance of FedFed, which intrinsically boils down to a relatively small $\sigma$ compared with sharing raw data $\mathbf{x}$.

## D.2 Analysis and Proof for Theorem 3.4

Previously, we showcase the view of each client for analyzing privacy. Albeit we do not aim at a new DP theorem, we expect a tighter privacy analysis for FedFed by using the advanced result [28]. Vadhan and Wang [61] prove that when the interactive mechanisms being composed are pure differentially private, their concurrent composition achieves privacy parameters (with respect to pure or approximate differential privacy) that match the (optimal) composition theorem for noninteractive differential privacy. To be specific, we follow the proof logic of Theorem D.7 [28] for analyzing globally shared data from all clients.

Given $(\epsilon_k, \delta)$-DP at each client side, we utilize the composition theorem to analyze overall privacy in FedFed. The concept of sensitivity below is originally used for sharing a dataset for achieving $(\epsilon, \delta)$-differential privacy.

**Definition D.5** (Sensitivity [28]). *The sensitivity of a query function $\mathcal{F} : \mathbb{D} \to \mathbb{R}$ for any two neighboring datasets $D, D'$ is $\Delta = \max_{D,D'} \|\mathcal{F}(D) - \mathcal{F}(D')\|$, where $\|\cdot\|$ denotes $L_1$ or $L_2$ norm.*

Privacy loss is a random variable that accumulates the random noise added to the algorithm/model, which is utilized in Theorem D.7.

**Definition D.6** (Privacy Loss [28]). *. Let $\mathcal{M} : \mathbb{D} \to \mathbb{R}$ be a randomized mechanism with input domain $D$ and range $R$. Let $D, D'$ be a pair of adjacent dataset and* aux *be an auxiliary input. For an outcome $o \in \mathbb{R}$, the privacy loss at $o$ is defined by $\mathcal{L}_{\mathsf{Pri}}^{(o)} \triangleq \log(\Pr[\mathcal{M}(\mathsf{aux}, D) = o]/\Pr[\mathcal{M}(\mathsf{aux}, D') = o])$.*

**Theorem D.7** (Composition Theorem [28]). *For any $\epsilon > 0, \delta \in [0, 1]$, and $\hat{\delta} \in [0, 1]$, the class of $(\epsilon, \delta)$-differentially private mechanisms satisfies $(\hat{\epsilon}_{\hat{\delta}}, 1 - (1 - \hat{\delta})\Pi_i(1 - \delta_i))$-differential privacy under $k$-fold adaptive composition for $\hat{\epsilon}_{\hat{\delta}} = \min\{k\epsilon, (e^\epsilon - 1)\epsilon k/(e^\epsilon + 1) + \epsilon\sqrt{2k \log(e + \sqrt{k\epsilon^2/\hat{\delta}})}, (e^\epsilon - 1)\epsilon k/(e^\epsilon + 1) + \epsilon\sqrt{2k \log(1/\hat{\delta})}\}$.*

*Proof.* Definition D.6 gets $\mathcal{L}_{\mathsf{Pri}}^{(o)}$ on an outcome variable $o$ over databases $D$ and $D'$. This proof starts with a particular random mechanism $\mathcal{M}^\dagger$ with further generalization. The mechanism $\mathcal{M}^\dagger$ does not depend on the database or the query but relies on hypothesis hp. For hp $= 0$, the outcome $O_i$ of $\mathcal{M}_i^\dagger$ is independent and identically distributed from a discrete random distribution $O^{\mathsf{hp}=0} \sim \mathcal{P}^{\dagger,0}$. $\mathcal{P}^{\dagger,0}(o)$ is defined to be: $\delta$ for $o = 0$; $(1 - \delta)e^\epsilon/(1 + e^\epsilon)$ for $o = 1$; $(1 - \delta)/(1 + e^\epsilon)$ for $o = 2$; $0$ for $o = 3$. For hp $= 1$, the outcome $O_i$ of $\mathcal{M}_i^\dagger$ is $O^{\mathsf{hp}=1} \sim \mathcal{P}^{\dagger,1}$. $\mathcal{P}^{\dagger,1}(o)$ is defined to be: $0$ for $o = 0$; $(1 - \delta)/(1 + e^\epsilon)$ for $o = 1$; $(1 - \delta)e^\epsilon/(1 + e^\epsilon)$ for $o = 2$; $\delta$ for $o = 3$.

Let $\mathcal{R}(\epsilon, \delta)$ be privacy region of a single access to $\mathcal{M}^\dagger$. The privacy region consists of two rejection regions with errors, i.e., rejecting true null-hypothesis (type-I error) and retaining false null-hypothesis (type-II error). Let $\epsilon_k^\dagger, \delta_k^\dagger$ be $\mathcal{M}_i^\dagger$'s parameters for defining privacy. $\mathcal{R}(\mathcal{M}, D, D')$ of any mechanism $\mathcal{M}$ can be regarded as an intersection of $\{(\epsilon_k^\dagger, \delta_k^\dagger)\}$ privacy regions. For an arbitrary mechanism $\mathcal{M}$, we need to compute its privacy region using the $(\epsilon_k^\dagger, \delta_k^\dagger)$ pairs. Let $D, D'$ be neighbouring databases and $\mathcal{O}$ be the outputting domain. Define (symmetric) $\mathcal{P}, \mathcal{P}'$ to be probability density function of the outputs $\mathcal{M}(D), \mathcal{M}(D')$, respectively. Assume a permutation $\pi$ over $\mathcal{O}$ such that $\mathcal{P}'(o) = \mathcal{P}(\pi(o))$.

Let $S$ denote the complement of a rejection region. Since $\mathcal{R}(\mathcal{M}, D, D')$ is convex, we have $1 - \mathcal{P}(S) \geq -e^{\epsilon_k^\dagger}\mathcal{P}'(S) + 1 - \delta_k^\dagger \Rightarrow \mathcal{P}(S) - e^{\epsilon_k^\dagger}\mathcal{P}'(S) \leq \delta_k^\dagger$. Define $\mathsf{Dt}_{\epsilon^\dagger}(\mathcal{P}, \mathcal{P}') = \max_{S \subseteq \mathcal{O}}\{\mathcal{P}(S) - e^{\epsilon^\dagger}\mathcal{P}'(S)\}$. Thus, $\mathcal{M}$'s privacy region is the set: $\{(\epsilon_k^\dagger, \delta_k^\dagger) : \epsilon_k^\dagger \in [0, \infty]\}$ s.t. $\mathcal{P}(o) = e^{\epsilon_k^\dagger}\mathcal{P}'(o), \delta_k^\dagger = \mathsf{Dt}_{\epsilon_k^\dagger}(\mathcal{P}, \mathcal{P}')\}$. Next, we consider composition on random mechanisms $\mathcal{M}_1, \ldots, \mathcal{M}_i$. By accessing $\mathcal{M}_i^\dagger$, $\mathcal{P}(O^{1,\mathsf{hp}} = o_1, \ldots, O^{i,\mathsf{hp}} = o_i) = \Pi_{j=1}^i \mathcal{P}^{\dagger,\mathsf{hp}}(o_j)$. By algebra on two discrete distributions, $\mathsf{Dt}_{(i-2j)\epsilon}(\mathcal{P}^i, (\mathcal{P}')^i) = 1 - (1 - \delta)^i + (1 - \delta)^i \sum_{l=0}^{j-1} \binom{i}{l}(e^{\epsilon(i-l)} - e^{\epsilon(i-2j+l)}))/(1 + e^\epsilon)^k$. Hence, privacy region is an interaction of $i$ regions, parameterized by $1 - (1 - \hat{\delta})\Pi_i(1 - \delta_i)$.

Now, by Lemma D.2, we get Theorem 3.4 directly. In summary, FedFed protects two types of data features using two different protective manners, i.e., small noise for performance-sensitive features and extremely large noise for performance-robust features, and thus attains higher model performance and stronger security in the same time.

## E   More Results on Attack

### E.1   More Details on Model Inversion Attack

In this section, we present additional results of the model inversion attack. When the central server assumes the role of the attacker without access to the original data, it exhibits a diminished performance in attacking the model. The raw data $\mathbf{x}$ of Figure 3 (a) and (b) can be found in Figure 6(a). As we can see, $\mathbf{x}_s$ still causes privacy leakage and Figure 6(b) also gives the intuitive necessity of DP protection on $\mathbf{x}_s$.

Inspired by generative model inversion methods [40, 62], we conduct another experiment where one of the clients acts as the attacker. Similar to GMI [40], the method involves leveraging an auxiliary dataset and a public dataset to launch an attack on the target model. In our experiments, we utilize the globally shared dataset as the auxiliary dataset, while the local private data serves as the public

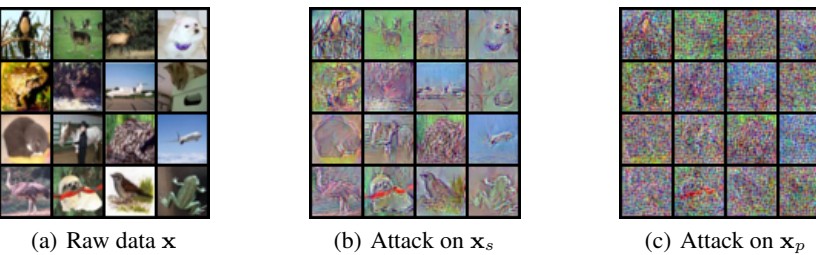

(a) Raw data $\mathbf{x}$            (b) Attack on $\mathbf{x}_s$            (c) Attack on $\mathbf{x}_p$

Figure 6: Model inversion attack. White-Box attack performance-sensitive features $\mathbf{x}_s$ and globally shared data $\mathbf{x}_p, \mathbf{x}_p = \mathbf{x}_s + \mathbf{n}, \mathbf{n} \sim \mathcal{N}(\mathbf{0}, \sigma_s^2 \mathbf{I})$.

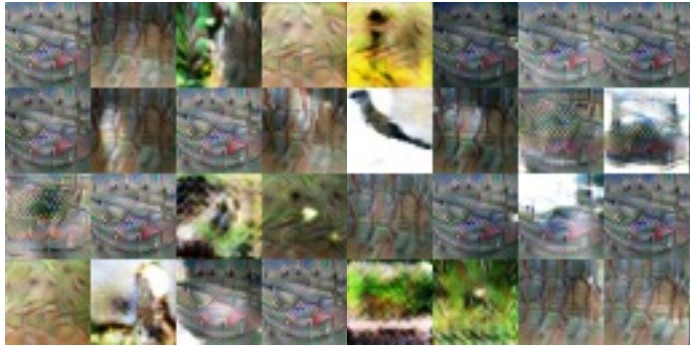

Figure 7: Generative-based model inversion attack results.

dataset because the client has access to its own local data. Under this scenario (results in Figure 7), the attacker's performance is better than when the server acts as the attacker, while the attacker is still unable to recover data from the shared features. The model architectures of generative-based model inversion attack are listed in Table 8.

### E.2 More Details on Membership Inference Attack

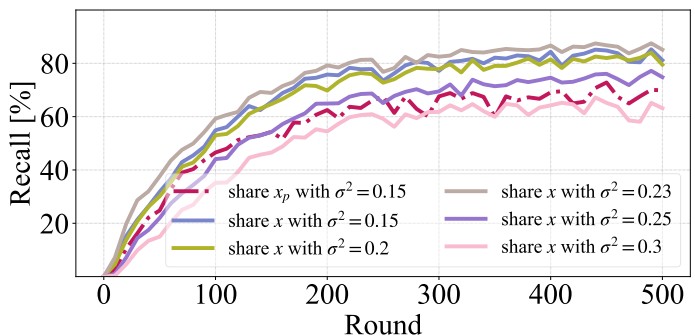

Figure 8: Attack process. Perform membership inference attack on global model per 10 communication rounds.

*Is Theorem 3.3 true empirically?*

Membership inference attack(MIA) [41, 15] attempts to infer whether a particular data point is in the target model's training set. Specifically, the global model on the server side can be regarded as the target model in FL. In our setting, we train a shadow model with globally shared data. Then the membership attack can be regarded as a binary classification task, member data or non-member data. The input of the attack model is the top-k vector of the output from the shadow model. To compare the superiority of sharing partial data rather than the complete data with DP, we conduct MIA to explore the divergence experimentally. Sharing raw data cause more information leakage (higher

Table 8: The model architectures used in generative-based model inversion attack

| The encoder structure of the generator takes as input the globally shared data | | | | |
|---|---|---|---|---|
| Type | Kernel | Dilation | Stride | Outputs |
| conv | 5x5 | 1 | 1x1 | 32 |
| conv | 3x3 | 1 | 2x2 | 64 |
| conv | 3x3 | 1 | 1x1 | 64 |
| conv | 3x3 | 1 | 2x2 | 128 |
| conv | 3x3 | 1 | 1x1 | 128 |
| conv | 3x3 | 1 | 1x1 | 128 |
| conv | 3x3 | 2 | 1x1 | 128 |
| conv | 3x3 | 4 | 1x1 | 128 |
| conv | 3x3 | 8 | 1x1 | 128 |
| conv | 3x3 | 16 | 1x1 | 128 |

| The encoder structure of the generator takes as input the latent vector | | | | |
|---|---|---|---|---|
| linear | | | | 2048 |
| deconv | 5x5 | | 1/2x1/2 | 256 |
| deconv | 5x5 | | 1/2x1/2 | 128 |

| The decoder structure of the generator | | | | |
|---|---|---|---|---|
| deconv | 5x5 | | 1/2x1/2 | 128 |
| deconv | 5x5 | | 1/2x1/2 | 64 |
| conv | 3x3 | | 1x1 | 32 |
| conv | 3x3 | | 1x1 | 3 |

| The global discriminator structure | | | | |
|---|---|---|---|---|
| conv | 3x3 | | 2x2 | 64 |
| conv | 3x3 | | 2x2 | 128 |
| conv | 3x3 | | 2x2 | 256 |
| conv | 3x3 | | 2x2 | 512 |
| conv | 1x1 | | 4x4 | 1 |

recall of MIA) than partial data with the same DP level. However, the FedFed needs a relatively small noise $\sigma$ to achieve comparable protection. The attack process is shown in Figure 8.

# F    Supplementary for Experiments

## F.1    Visualization of Data Heterogeneity

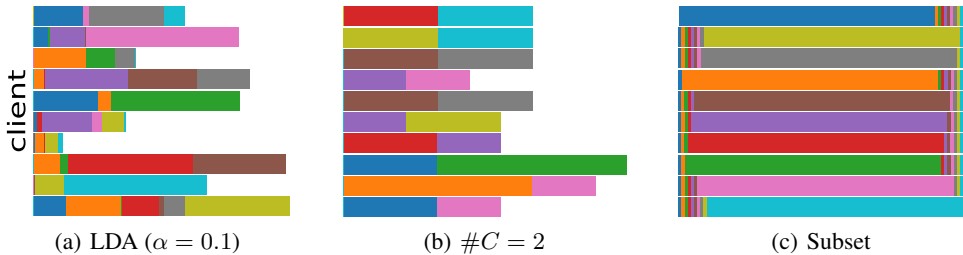

(a) LDA ($\alpha = 0.1$)  (b) $\#C = 2$  (c) Subset

Figure 9: Data distribution in various FL heterogeneity scenarios. Different colours denote different labels and the length of each line denotes the data number. As we can see, in our FL setting, we mainly perform on two kinds of Non-IID scenarios, including label skew and quantity skew.

We show the visualization of data distribution in Figure 9. The LDA partition and the $\#C = 2$ partition have the label skew and the quantity skew simultaneously. And the Subset partition only has

the label skew. $\#C = k$ means each client only has $k$ different labels from dataset, and $k$ controls the unbalanced degree. The subset method makes each client have all classes from the data, but one dominant class far away outnumbers other classes.

## F.2   More Guessing Games

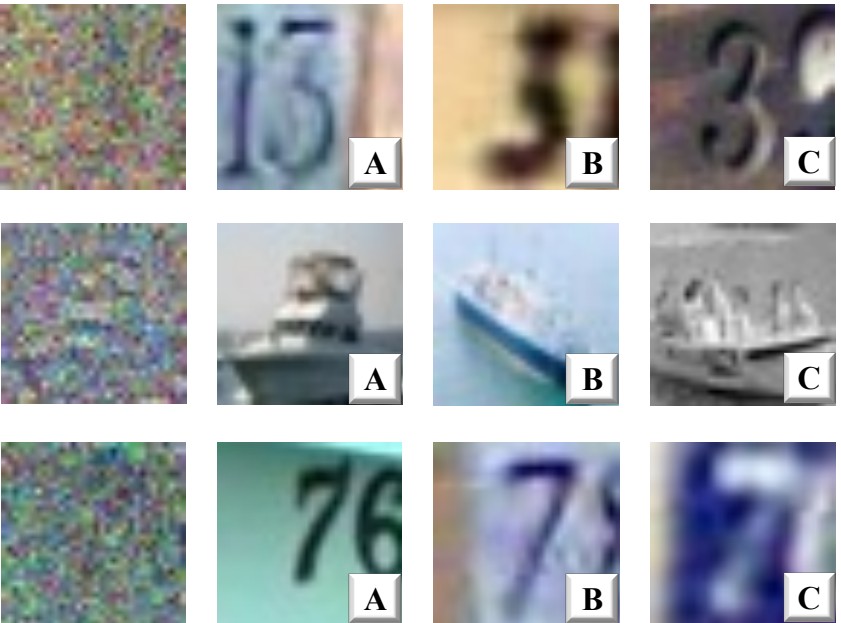

Figure 10: Guessing Games
Question: Which one is the first image from in each row? A or B or C?

Let's play more guessing games here. We selected samples from the same category for the purpose of reducing the difficulty. The answer in Figure 1 is C. The answers in Figure 10 are (B, A, A) in order.

## F.3   Sharing Protected Partial Features vs. Protected Raw Data

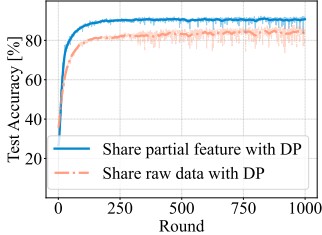

Figure 11: Performance on different information sharing strategies with DP protection: raw data and $\mathbf{x}_p$ under the same protection strength.

In this section, we compare the performance divergence of applying DP to protect different sharing information, i.e., performance-sensitive features $\mathbf{x}_p$ and raw data $\mathbf{x}$. The results is shown in Figure 11

## F.4   More experimental results

We provide more comparison results in this section to demonstrate the superiority of FedFed. Figure 12, Figure 13, Figure 14, Figure 15 have shown our enhancement in four datasets. In this paper, we mainly have four settings in our experiments: (1) $\alpha = 0.1, E = 1, K = 10$; (2) $\alpha = 0.1, E = 5, K = 10$; (3) $\alpha = 0.05, E = 1, K = 10$; (4) $\alpha = 0.1, E = 1, K = 100$.

Table 9: Different sampling rates of FedFed under $\alpha = 0.1, E = 1, K = 100$ over CIFAR-10

| Sampling rates | 5% | 10% | 20% | 40% | 60% |
|---|---|---|---|---|---|
| Accuracy | 83.67 | 84.06 | 87.98 | 89.17 | 89.62 |
| Round to reach target accuracy | 182 | 163 | 90 | 70 | 61 |

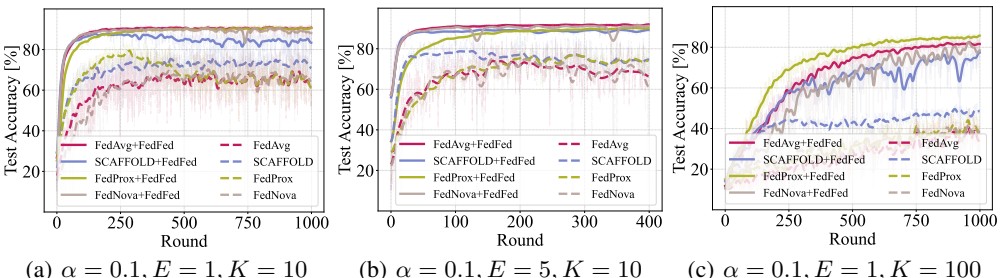

(a) $\alpha = 0.1, E = 1, K = 10$    (b) $\alpha = 0.1, E = 5, K = 10$    (c) $\alpha = 0.1, E = 1, K = 100$

Figure 12: Convergence and test accuracy comparison on CIFAR-10.

In addition, we investigate the impact of varying sampling rates on the performance of our FL system, and present the results in Table 9. We observe that increasing the sampling proportion leads to gradual improvements in the model's performance, as well as faster convergence. This is due to a larger number of clients participating in each round, resulting in more consistent update directions for the aggregated model and the global model.

We also conduct tests on FedFed using various levels of heterogeneity, such as $\alpha = 0.5$ and $\alpha = 1$. The results are presented in Tables 10 and 11. As the value of $\alpha$ increases, the original performance of FedAvg demonstrates significant improvement, while the performance enhancement of FedAvg deployed with FedFed is comparatively lower. This discrepancy can be attributed to the reduction in heterogeneity as $\alpha$ increases. When $\alpha$ reaches 1.0, the data distribution among clients becomes nearly homogeneous. As a result, the effectiveness of FedFed in mitigating heterogeneity through information sharing diminishes significantly.

### F.5 More Results on Impact of DP noise

In this section, we present additional results on the impact of DP noise on three other datasets, as shown in Figure 16. Our findings indicate that as the level of DP noise increases, the privacy budget decreases and the ability to protect information increases, but the performance of the model decreases accordingly. Thus, it is crucial to find a balance between privacy and performance in practical applications of our FedFed.

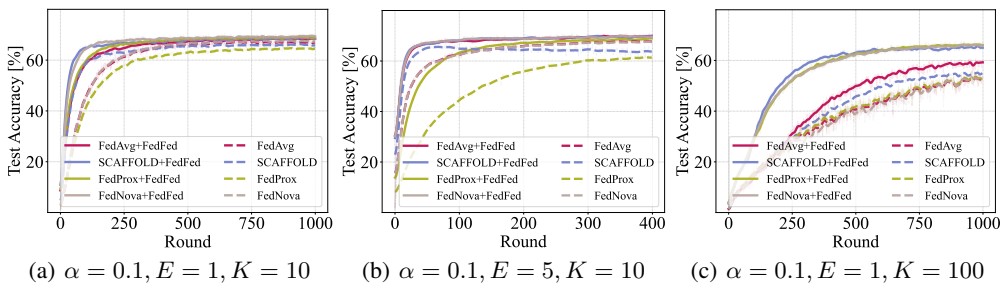

(a) $\alpha = 0.1, E = 1, K = 10$    (b) $\alpha = 0.1, E = 5, K = 10$    (c) $\alpha = 0.1, E = 1, K = 100$

Figure 13: Convergence and test accuracy comparison on CIFAR-100.

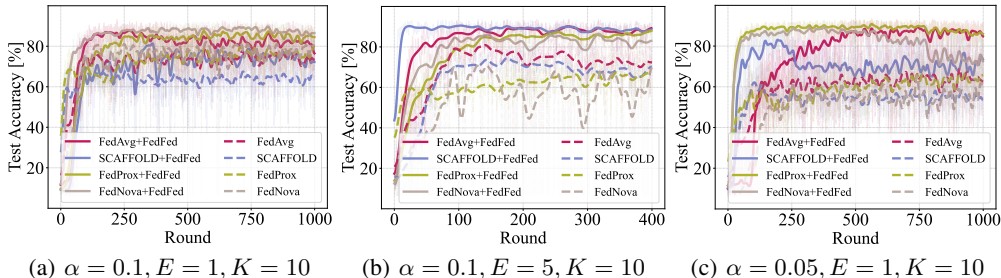

Figure 14: Convergence and test accuracy comparison on SVHN.

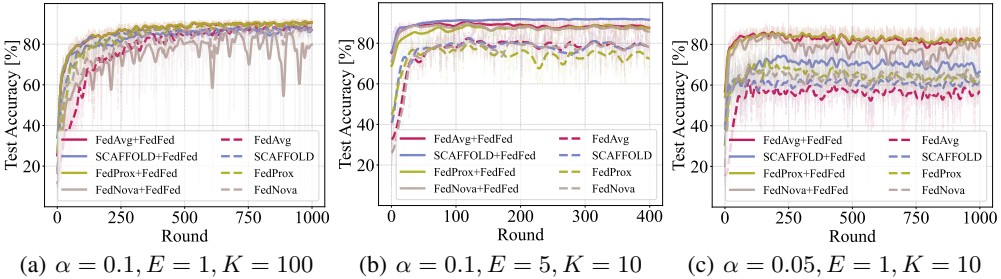

Figure 15: Convergence and test accuracy comparison on FMNIST.

## F.6 More Explanation: Performance-Sensitive Features vs. Performance-Robust Features

For an intuitive understanding of why there is no drastic performance degradation for utilizing $\mathbf{x}_s$ as a substitute for raw data $\mathbf{x}$. Formally, e.g. in the classification task, a classifier trained by $\mathbf{x}$ gets comparable test accuracy on $\mathbf{x}_s$ giving the verification that $\mathbf{x}_s$ contains the primary features for generalization and vice versa. Specifically, three classifiers trained on $\mathbf{x}$, $\mathbf{x}_s$, and $\mathbf{x}_r$, Then test the generalization ability on $\mathbf{x}$, $\mathbf{x}_r$, and $\mathbf{x}_s$, separately. The result is shown in Figure 17.

As described above, $\mathbf{x}_r$ extract from $\mathbf{x}$ in an IB manner contains more visual information about private data $\mathbf{x}$ than $\mathbf{x}_s$. To measure the visual similarity among performance-robust features $\mathbf{x}_r$, performance-sensitive features $\mathbf{x}_s$, and raw data $\mathbf{x}$. We choose PSNR [63, 64], prominent visual information comparisons of images, to calculate pixels error, the result is shown in Table 12.

Consequently, higher PSNR denotes that $\mathbf{x}_r$ has the most visual information of $\mathbf{x}$ than $\mathbf{x}_s$. While $\mathbf{x}_r$ shares similarities with $\mathbf{x}$ in terms of its form or structure, it exhibits limitations that hinder its ability to enhance performance. At the same time, $\mathbf{x}_s$ has the commensurate generalization ability like $\mathbf{x}$ to achieve our goal.

## F.7 Overheads Analysis of FedFed

For the extra computation induced by FedFed, we provide two views to demonstrate the limited costs of extra computation, i.e., training time and FLOPs.

Table 13 shows that training a generator requires less than $10\%$ of the time needed to train a classifier. Therefore, the additional training time required for the generator is limited.

Table 10: Top-1 Accuracy of $\alpha = 0.5, E = 1, K = 10$ with $50\%$ sampling rate.

|  | CIFAR-10 | FMNIST | SVHN | CIFAR-100 |
|---|---|---|---|---|
| FedAvg | 87.68 | 90.32 | 91.11 | 68.98 |
| FedFed (Ours) | 93.21 | 94.01 | 93.71 | 69.52 |

Table 11: Top-1 Accuracy of $\alpha = 1.0, E = 1, K = 10$ with $50\%$ sampling rate.

|  | CIFAR-10 | FMNIST | SVHN | CIFAR-100 |
|---|---|---|---|---|
| FedAvg | 89.45 | 92.38 | 92.03 | 69.02 |
| FedFed (Ours) | 94.01 | 94.31 | 93.89 | 70.31 |

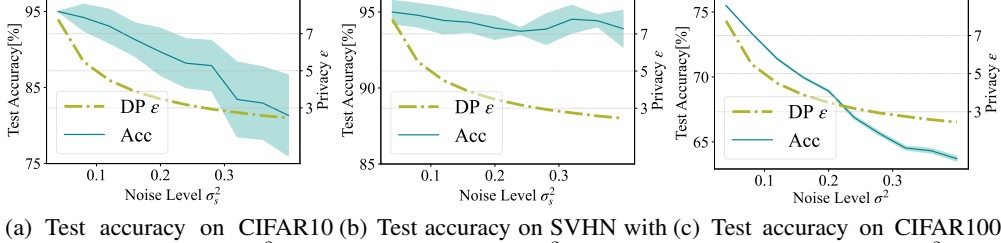

(a) Test accuracy on CIFAR10 with different noise level $\sigma_s^2$

(b) Test accuracy on SVHN with different noise level $\sigma_s^2$

(c) Test accuracy on CIFAR100 with different noise level $\sigma_s^2$

Figure 16: The impact of DP noise on different datasets.

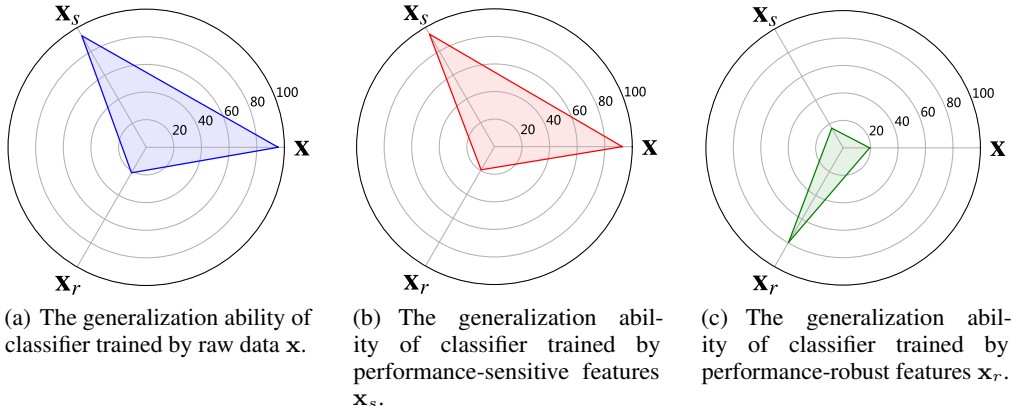

(a) The generalization ability of classifier trained by raw data $\mathbf{x}$.

(b) The generalization ability of classifier trained by performance-sensitive features $\mathbf{x}_s$.

(c) The generalization ability of classifier trained by performance-robust features $\mathbf{x}_r$.

Figure 17: The generalization ability graphs of three classifiers trained by different data types. The generalization ability means a well-trained model tests unseen information and measures the capability to finish a certain task, i.e., classification test accuracy. In our setting, we evaluate the generalization ability of classifiers on different information forms: raw data $\mathbf{x}$, performance-sensitive features $\mathbf{x}_s$, performance-robust features $\mathbf{x}_r$.

Table 12: The visual similarity of various feature types. PSNR of $\mathbf{x}$ and $\mathbf{x}_r$, $\mathbf{x}$ and $\mathbf{x}_s$.

|  | $\mathbf{x}_r$ | $\mathbf{x}_s$ |
|---|---|---|
| $\mathbf{x}$ | $31.47 \pm 1.57$ | $18.87 \pm 0.98$ |

Tables 15 and 14 show the model complexity results, which indicate that the computation required for the generator is approximately 8.2% of that required for the classifier. Additionally, the number of communication rounds needed for the generator is significantly lower than that required by the classifier, with only 15 rounds compared to 1000 rounds for the classifier (namely, $T_d = 15$ and $T_r = 1000$).

The use of FedFed incurs additional memory usage, as the size of the globally-shared dataset is equivalent to the combined size of all clients' data. This may have limited impacts on the bandwidth, as the generated data is only sent once. To quantitatively measure its effects, we compared the size of the dataset with that of the classifiers in Table 16. Sharing the generated dataset is equivalent to sharing classifiers in multiple rounds. For example, for CIFAR10, the data requires approximately 586MB of memory, while the classifier requires about 48MB. Thus, sharing the global dataset requires the same amount of communication as sending a classifier in approximately 14 communication rounds.

Based on the aforementioned analysis, we provide a comprehensive examination of the communication overheads associated with the FedFed. Consider $K$ clients in the FL system. Let $m$ be the size of a single local model. The size of local private data is $\|\mathcal{D}_k\| = a$. Then, the ratio of the entire dataset to a model parameter is $\gamma$, where $\gamma = \frac{aK}{m} \approx 14$. The extra communication cost for a single client is $(m + m) * T_d + a + aK$ where $(m + m) * T_d$ denotes the cost of download/upload models for $T_d$ rounds in Algorithm 1. Here, the $a$ denotes the performance-sensitive features sent by each client, and $aK$ is the data received by each client from the globally shared dataset. In the general process of FL, the overall communication costs are $(m + m)T_r\beta$, where $\beta$ is the sampling rate of a client. Therefore, the ratio of the extra communication overhead to the general FL process is:

$$\frac{(m + m) \cdot T_d + a(K + 1)}{(m + m) \cdot T_r \cdot \beta} = \frac{T_d}{T_r \cdot \beta} + \frac{a(K + 1)}{2m \cdot T_r \cdot \beta} = \frac{T_d}{T_r \cdot \beta} + \frac{\gamma}{2T_r \cdot \beta} + \frac{\gamma}{2K \cdot T_r \cdot \beta} \quad (17)$$

Here, we detail two examples in our experiments:

* For $K = 10, T_d = 15, T_r = 1000$, and $\beta = 50\%$, the extra communication costs are approximately 4.54%.
* When $K = 100, T_d = 15, T_r = 1000$, and $\beta = 10\%$, the extra communication costs are approximately 22.07%.

To facilitate the further deployment of FedFed, we offer three strategies tailored to different storage hardware configurations:

1. One-time download: Local clients download the globally shared dataset once. A globally shared dataset costs approximately $14\times$ the storage of a classifier model.

2. Partial download: A small portion of the globally shared dataset is selected and downloaded. This strategy incurs approximately $1.5\times$ communication cost compared to the previous strategy, while the storage required by the clients is the same as that of local private data. This represents a storage-friendly choice that may involve a trade-off in terms of performance.

3. Intermittent download: A small set of globally shared dataset is downloaded after every $Z$ rounds. This approach reduces the communication overhead to $\frac{1}{Z}$ of that of strategy 2 while maintaining the storage overhead at the size of the local data.

Table 13: Comparison of training time between Classifier and Generator

| | |
|---|---|
| Training time of Generator | 5,251 s |
| Training time of Classifier | 62,901 s |

Table 14: Parameters of Classifier and Generator

| | |
|---|---|
| Parameters of Generator | 48M |
| Parameters of Classifier | 42MB |

# G  More Related Works

## G.1  Federated Learning with Heterogeneous Data.

FL allows the distributed clients to train a model across multiple datasets jointly. It "protects" user privacy by controlling data accessibility among different clients, i.e., only the data owner has the right

Table 15: FLOPs of Classifier and Generator

| | |
|---|---|
| FLOPs of training Generator | 16.03M FLOPs |
| FLOPs of training Classifier | 556.66M FLOPs |

Table 16: Parameters of Local Classifier and Globally-shared Data

| | |
|---|---|
| Parameters of Local Classifier | 42MB |
| Parameters of Globally-shared Data | 586MB |

Table 17: Existing method about information sharing strategy to mitigate data heterogeneity in Federated Learning.

| | Shared Info | Protection on Shared Info | Attack Test |
|---|---|---|---|
| FD+FAug [46] | Model output & Synthetic info | ✗ | — |
| FedMD [65] | logits | ✗ | — |
| XorMixFL [48] | Statistic of data | ✗ | — |
| FedDF [66] | Statistic of logits | ✗ | — |
| FedProto [50] | Abstract class prototypes | ✗ | — |
| FedFTG [67] | Synthetic info | ✗ | — |
| CCVR [10] | Statistic of logits | ✗ | Model inversion attack |
| Fed-ZDAC [12] | Batch normalization features | ✔ | — |
| FedAUX [68] | Synthetic info | ✔ | — |
| **FedFed (ours)** | Protected partial features | ✔ | Model inversion attack
Membership inference attack |

to access the corresponding data. FedAvg [4] is the seminal work designed to reduce communication via more local training epochs and fewer communication rounds. In the following, many works [9, 5] observe that the FedAvg's divergence is considerable compared with centralized training if clients hold heterogeneous distribution. Even worse, the gap accumulates as the weight aggregates, potentially hurting model performance.

Recently, a series of works have tried to calibrate the updated direction of local training from the global model. FedProx [6] adds an $L_2$ distance as the regularization term in the objective function, providing a theoretical guarantee of convergence. Similarly, FedIR [42] operates on a mini-batch by self-normalized weights to address the non-identical class distribution. SCAFFOLD [7] restricts the model using the previous knowledge. Besides, MOON [43] introduces contrastive learning at the model level to correct the divergence between clients and the server.

Meanwhile, recent works propose designing new model aggregation schemes. FedAvgM [33] performs momentum on the server side. FedNova [22] adopts a normalized averaging method to eliminate objective inconsistency. A study [69] also indicates that biasing client selection with higher local loss can speed up the convergence rate. The coordinate-wise averaging of weights also induces noxious performance. FedMA [44] conducts a Bayesian non-parametric strategy for heterogeneous data. FedBN [45] focuses on feature shift Non-IID and performs local batch normalization before averaging models.

Another existing direction for tackling data heterogeneity is sharing data. This line of work mainly assembles the data of different clients to construct a global IID dataset, mitigating client drift by replenishing the lack of information of clients [9]. Existing methods include synthesizing data based on the raw data by GAN [46]. However, the synthetic data is generally relatively similar to the raw data, leading to privacy leakage at some degree. Adding noise to the shared data is another promising strategy [70, 71]. Some methods employ the statistics of data [48] to synthesize for sharing, which still contains some raw data content. Other methods distribute intermediate features [12], logits [49, 10], or learn the new embedding [50]. These tactics will increase the difficulty of privacy protection because some existing methods can reconstruct images based on feature inversion methods [51]. Most of the above methods share information without a privacy guarantee or with strong privacy-preserving but poor performance, posing the privacy-performance dilemma.

Concretely, in FD [46] all clients leverage a generative model collaboratively for data generation in a homogeneous distribution. For better privacy protection, G-PATE [47] performs discriminators with local aggregation in GAN. Fed-ZDAC(Fed-ZDAS) [12], depending on which side to play augmentation, introduce zero-shot data augmentation by gathering intermediate activations and batch

normalization(BN) statistics to generate fake data. Inspired by mixup data. Cronus [49] transmits the logits information while CCVR [10] collects statistical information of logits to sample fake data. FedFTG [67] use a generator to explore the input space of the local model and transfer local knowledge to the global model. FedDF [66] utilizes knowledge distillation based on unlabeled data or a generator and then conducts *AVGLOGITS*. We summarize existing information-sharing methods in Table 17 to compare the dissimilarity with FedFed. The main difference between FedDF and FedFed is that our method distils raw data into two parts (performance-sensitive features $x_e$ and performance-robust features $x_r$) rather than transferring distilled knowledge. We provide *hierarchical protections* to preserve information privacy while overcoming the privacy-performance dilemma.

## G.2 Differential Privacy with Federated Learning

Carlini et.al [72] found that memorizing sensitive data occurs in early training, regardless of data rarity and model.

Training with differential privacy [73, 74] is a feasible solution to avoid its risk, albeit at some loss in utility. Differential privacy guarantees that an adversary should not discern whether a client's data was used.

Huang et al [75] and Wei et al [76] are the first (to their knowledge) to analyze the relation between convergence and utility in FL. Andrew et al [77] explore setting an adaptive clipping norm in the federated setting rather than using a fixed one. Andrew et al [77] explore setting an adaptive clipping norm in the federated setting rather than using a fixed one. They show that adaptive clipping to gradients can perform as well as any fixed clip chosen by hand. Hoeven et al [78] introduce data-dependent bounds and apply symmetric noise in online learning, which allows data providers to pick noise distribution. Sun et al [79] explicitly vary ranges of weights at different layers in a DNN and shuffle high-dimensional parameters at an aggregation for easing explodes of privacy budgets. Peng et al [80] study the knowledge embedding problem using DP protection in FL. Applying differential privacy and its variants to the federated setting become more prevailing nowadays.

