# OpenReview forum: "FedFed: Feature Distillation against Data Heterogeneity in Federated Learning"
_NeurIPS.cc/2023/Conference — NeurIPS 2023 poster_

### Official Review · Reviewer_ezSw · 2023-06-20

**Soundness:** 2 fair
**Presentation:** 3 good
**Contribution:** 2 fair
**Rating:** 5
**Confidence:** 4

**Summary:**

This paper proposes a novel approach called Federated Feature Distillation (FedFed) to mitigate the data heterogeneity problem while preserving privacy. In particular, FedFed partitions data into performance-sensitive features and performance-robust features based on the information bottleneck method. Only performance-sensitive features are shared among clients as they contain minimal private information and significantly contribute to performance. Moreover, incorporating the differential privacy (DP) mechanism can provide an additional layer of privacy protection. In summary, this work is interesting, and the authors provide some theoretical analyses to support their claims.

**Strengths:**

This study mitigates the issue of data heterogeneity in Federated Learning (FL) by utilizing a promising approach to information-sharing. Empirical evidence shows that the proposed method is effective in enhancing model performance.

**Weaknesses:**

The reviewer believes the evaluation of the privacy leakage is insufficient. The details are as follows:

1. In Section 4.4, the authors conducted a model inversion attack to infer private data using shared features. However, quantitative measurement is missing in this evaluation. Previous studies [1, 2] have utilized metrics such as peak signal-to-noise ratio (PSNR) and Frechet inception distance (FID) to assess the quality of reconstructed data. We recommend that the authors include some quantitative results to strengthen the empirical evidence.

2. Federated learning is known to leak private information when sharing model parameters [3,4]. The FedFed method, which shares both model parameters and features, creates an opportunity for attackers to exploit these two types of information for privacy attacks. However, the experiment showed that the attacks only targeted the privacy information contained in the features, which is not a comprehensive representation of potential privacy leaks.

3. Tables 1, 2, 3, and 4 demonstrate that the FedFed method achieves a higher Top-1 accuracy than FedAvg. However, it's worth considering whether this increased accuracy comes at the cost of higher privacy leakage.

[1] The Secret Revealer: Generative Model-Inversion Attacks Against Deep Neural
Networks. https://openaccess.thecvf.com/content_CVPR_2020/html/Zhang_The_Secret_Revealer_Generative_Model-Inversion_Attacks_Against_Deep_Neural_Networks_CVPR_2020_paper.html

[2] Knowledge-Enriched Distributional Model Inversion Attacks. https://openaccess.thecvf.com/content/ICCV2021/html/Chen_Knowledge-Enriched_Distributional_Model_Inversion_Attacks_ICCV_2021_paper.html

[3] Comprehensive Privacy Analysis of Deep Learning: Passive and Active White-box Inference Attacks against Centralized and Federated Learning. https://ieeexplore.ieee.org/abstract/document/8835245?casa_token=1iwvBNyN5q4AAAAA:BdxQzounj3eoNv0HIcdMoW7nCaM6xWJFPwZQIosqhvpiXWNaJd-q0MeW_xmiZJkZGVmZXDRE

[4] Beyond Inferring Class Representatives: User-Level Privacy Leakage From Federated Learning. https://ieeexplore.ieee.org/abstract/document/8737416?casa_token=zyEVcT_x3oQAAAAA:FfpfqLHOsQJ33Br2OqnYE6fRI3EIgdNPlCSUOC74Yu6qhFjcgJvqHwFLAsPaShPaigK3sjz1


**Questions:**

1. How to conduct the model inversion attack? The reviewer has checked both the experiment section and Appendix E, but some details appear to be missing.

2. Table 14 in the Appendix summarizes the currently available information-sharing methods. According to the authors, the FedFed method provides hierarchical protections to preserve information privacy while overcoming the privacy-performance dilemma. Does this mean that the FedFed method achieves better performance and privacy protection than all the baselines in Table 14? Take Fedproto as an example. This method only exchanges prototypes (i.e., the mean of features) instead of model parameters, while FedFed shares both the model parameters and features. The authors correctly pointed out that FedProto lacks protection on the shared information, but the shared model parameters in FedFed also lack privacy guarantees. Further discussion and empirical results are required to demonstrate the effectiveness of the FedFed method in comparison to previous works.

**Limitations:**

The authors correctly pointed out that the performance of FedFed is limited by the storage capacity of clients.

---

> ### Author Rebuttal · Authors · 2023-08-10
>
> ## Response to Reviewer ezSw:
> > Q1:Quantitative measurement is missing in the evaluation of model inversion attack in Sec.4.4, e.g., PSNR and FID.
>
> **A1:** Thank you for your valuable suggestion. Accordingly, we employ PSNR and FID, used in [R1][R2], to provide quantitative results for the samples reconstructed by model inversion attack.  We will add these results to our revision.
> - As in Table 1, if the central server acts as the attacker, it does not have access to the original data, resulting in a poorer performance in attacking the model.
> - Inspired by your comments, we conducted another experiment where one of the clients acts as the attacker. The client has access to its local data. Under this scenario, the attacker's performance is better than when the server acts as the attacker, while the attacker is still unable to recover data from the shared features.
>
> Table 1. The PSNR and FID comparison under two kinds of attack methods.
> |  | PSNR| FID|
> |---|---|---|
> | Server acts as a malicious attacker | 15.56 |404.16|
> | One of the clients acts as a malicious attacker | 18.31 |378.21|
> > Q2: How to conduct the model inversion attack?
>
> **A2:** We assume that the server is a malicious attacker, i.e., the server can access the globally shared data and the global model to conduct a model inversion attack. The results reported in Sec. 4.4 (PSNR is 15.56).
> Following the valuable comments, we conduct another model inversion attack introduced in [R1]. Specifically, the attacker (one client in the FL system) uses a public dataset (the local data of the malicious client) and auxiliary data (the shared features of FedFed) to train a GAN to attack target models, namely, the GAN used to reconstruct other clients' private data. In our experiments, the attacker hardly recovers others' private data, where the PSNR of the recovered results is 18.31.
>
> > Q3:  The experiment showed that the attacks only targeted the privacy information contained in the features, which is not a comprehensive representation of potential privacy leaks.
>
> **A3:** We agree that sharing model parameters may risk privacy. In this regard, sharing more information could creates an opportunity for attackers to exploit these two types of information for privacy attacks.
> - Consequently, as mentioned in the comments, we conducted privacy attacks to show that attackers hardly get information in the shared features.
> - Until now, our work has explored the most sufficient attacks compared with previous works [R3-R4], which employ an information-sharing strategy to tackle data heterogeneity in FL.
> We totally agree with your comments that a comprehensive analysis of privacy attacks is promising and important.
> - Actually, we would like to remind you that it seems to be out-of-scope for a 9-page conference paper since our work aims at a new approach to tackle data heterogeneity. We prioritize mitigating the dilemma introduced by the information-sharing strategy.
> - Moreover, we believe that the reviewer may look forward to more work on investigating privacy attacks in FL, for those with the information-sharing strategy. In this regard, we sincerely expect that FedFed could serve as a baseline model for conducting privacy/attack research in the future. Also, the authors hope to contribute to the deep learning community more.
>
> > Q4:Tables 1-4 demonstrate that the FedFed method achieves a higher Top-1 accuracy than FedAvg. However, it's worth considering whether this increased accuracy comes at the cost of higher privacy leakage.
>
> **A4:** Sharing information of clients will inevitably introduce some privacy leaks, e.g., sharing models or features. Thus, FedFed distils performance-sensitive features and shares these features with DP protection to mitigate privacy leakage. We will add the discussion in our revision.
>
> > Q5: Does Table 14 mean that the FedFed method achieves better performance and privacy protection than all the baselines in Table 14? Further discussion and empirical results are required to demonstrate the effectiveness of the FedFed method in comparison to previous works.
>
> **A5:** We appreciate your valuable time and detailed comments. We will add further discussion in our revision.
> - We listed many excellent works in the research of the information-sharing strategy in Table 14, aiming to highlight the difference between FedFed and these works. Specifically, FedFed shares information with protection, while previous works do not protect the shared information. Thus, it is not fair for previous works to make a direct comparison. This is the reason why we do not report the comparison with these methods.
> - When talking about privacy and security, we need to consider the attacker's ability and privacy goals. Empirically, we have conducted multiple attacking experiments to explore and verify the improved protection in different settings.
> - Besides, analyzing all existing attacks with a sufficient theoretical guarantee for shared FL models is out-of-scope for a 9-page conference paper. The shared features are protected with DP. For the mentioned both shared parameters and features, we agree that studying how to perform an attack is a promising direction for FL security. However, our work focuses on tackling data heterogeneity through an information-sharing strategy with a DP guarantee.
> Thanks for the valuable comments, we will investigate more attacks induced by FL with the information-sharing strategy in our future work.
>
> **Reference:**\
> [R1] Zhang Y, Jia R, Pei H, et al. The secret revealer: Generative model-inversion attacks against deep neural networks. In CVPR, 2020.\
> [R2] Chen S, Kahla M, Jia R, et al. Knowledge-enriched distributional model inversion attacks. In CVPR, 2021.\
> [R3] Li D, Wang J. Fedmd: Heterogenous federated learning via model distillation. arXiv preprint arXiv:1910.03581.\
> [R4] Lin T, Kong L, Stich S U, et al. Ensemble distillation for robust model fusion in federated learning. In NeurIPS, 2020.

---

> > ### Comment · Reviewer_ezSw · 2023-08-14
> >
> > Thanks for the detailed answers to my comments. This addresses a few of my concerns, and as a result, I have increased my score from 4 to 5.
> >
> > However, the reviewer remains concerned that sharing partial features in data compromises privacy, and thus it is important to compare the privacy-performance tradeoff of the proposed method with the baselines.
> >
> > The reviewer understands that providing a comprehensive analysis of privacy attacks is beyond the scope of this paper. But it is recommended to highlight the additional privacy leakage as a limitation in the manuscript.

---

> > > ### Author Response · Authors · 2023-08-14
> > > **Response to ezSw**
> > >
> > > Thanks for your further suggestions. We agree with you that it is important to compare the privacy-performance trade-off of the existing methods. We will highlight the additional privacy leakage in the updated revision. Thank you for dedicating your time to our paper and raising the score.

---

### Official Review · Reviewer_wKGY · 2023-07-02

**Soundness:** 2 fair
**Presentation:** 2 fair
**Contribution:** 3 good
**Rating:** 7
**Confidence:** 4

**Summary:**

The paper proposes a method based on feature distillation to tackle data heterogeneity. The main contribution , as I see it, is in identifying performance-robust and performance-sensitive features and sharing the latter among clients to mitigate the impact of heterogeneity.

**Strengths:**

- The method is simple (this is very important) and can be easily used with existing FL algorithms.
- DP is incorporated to protect the leakage of privacy-sensitive information when sharing performance-sensitive features.

**Weaknesses:**

- The evaluation is quite limited to vision datasets, which makes me unsure if it is going to generalize across modalities.
- Number of clients (K) are also quite limited. In real-world setting 100-1000s of clients are involved in FL process.


**Questions:**

Feature distillation idea is proposed in [1]. Isn't the idea same? Please cite [1].
[1] Romero, A., Ballas, N., Kahou, S. E., Chassang, A., Gatta, C., & Bengio, Y. (2014). Fitnets: Hints for thin deep nets. arXiv preprint arXiv:1412.6550.

**Limitations:**

- Do not see discussion on whether performance-sensitive features indeed leak information.
- Formal definitions of performance-robust and performance-sensitive features are missing. Please also make a subsection to provide detailed descriptions of how these two types of features are identified.
- The experiments are only performed using ResNet-18.

---

> ### Author Rebuttal · Authors · 2023-08-09
>
> ## Response to Reviewer wKGY:
>
> > Q1. The evaluation is quite limited to vision datasets, which makes me unsure if it is going to generalize across modalities.
>
> **A1:** Thanks for your insightful comments. In this work, we merely focus on the image modality, while omitting the results on other modalities, e.g., text data. This is because applying information bottleneck (the core of FedFed) to text data requires a special design. In response to your valuable comments, we will explore approaches to applying FedFed to text data.
>
>
> > Q2: The number of clients (K) is also quite limited. In real-world settings, 100-1000s of clients are involved in the FL process.
>
> **A2:** Thanks for your valuable suggestion.
> - We followed the experimental settings used in our experiments, omitting results evaluated using more clients.
> - In response to your valuable comments, we conduct experiments using more clients, i.e., a more realistic setting. We test $K=500$ and $K=1000$ under $\alpha=0.1$ and the sampling rate is 5%, the results are listed in Tables 1 and 2. The results show that FedFed can improve the performance of the baseline method under a more realistic setting. We will add the results to our revision.
>
>  Table 1. The accuracy of $\alpha=0.1, E = 1, K= 500$  over various datasets.
> |Method|CIFAR-10 |FMNIST|SVHN|CIFAR-100|
> |:-----:|:-----:|:-----:|:-----:|:-----:|
> |FedAvg|47.15%|85.34%|86.31%|45.31%|
> |FedFed(Ours)|**81.67%**|**87.63%**|**87.56%**|**57.78%**|
>
>  Table 2. The accuracy of $\alpha=0.1, E = 1, K= 1000$  over various datasets.
> |Method|CIFAR-10 |FMNIST|SVHN|CIFAR-100|
> |:-----:|:-----:|:-----:|:-----:|:-----:|
> |FedAvg|43.15%|81.34%|82.31%|41.31%|
> |FedFed(Ours)|**78.36%**|**83.63%**|**84.56%**|**52.78%**|
>
> > Q3: Do not see a discussion on whether performance-sensitive features indeed leak information.
>
> **A3:** We will add the discussion in our revision.
> - Theoretically. The sharing strategy will leak information, but FedFed employs differential privacy to protect the shared feature. Thus, FedFed can share information with a given privacy budget, theoretically guaranteed with DP theory.
> - Empirically. We conduct privacy verification experiments including model inversion attack and membership inference attack in Sec 4.4. Besides, we perform an additional model inversion attack. The results are reported in the response to Reviewer #ezSw.
>
> > Q4: Formal definitions of performance-robust and performance-sensitive features are missing. Please also make a subsection to provide detailed descriptions of how these two types of features are identified.
>
> **A4:** Thanks for your kind suggestions. We will add the following descriptions to our revision.
> FedFed is built upon the information bottleneck theory, leading to the decision of performance-sensitive and performance-robust features. Specifically, performance-sensitive features $X_s$ statisfy that: $I(Y;X|X_s)=0$, namely, $X_s$ contains all information about the label $Y$. In contrast, the performance-rbust features $X_r$ contains all information about $X$ except for information about label $Y$, i.e., $I(Y;X_r)=0, I(X_r;X)=I(X_r;X|X_s)$. That is, $X_r$ contains all the left information about $X$ when removing $X_s$ from $X$. Thus, we define the performance-robust feature $X_r$ as the difference, i.e., $X_r = X - X_s$, and propose Eq.(5) to realize the intuition.
>
> > Q5: The experiments are only performed using ResNet-18.
>
> **A5:** Thanks for the valuable suggestions, motivating us to conduct additional experiments on two different model architectures.
> As reported in Tables 3-4, FedFed can improve model performance under various model architectures.
>
> Table 3. The comparison of our methods on various datasets with the VGG classifier.
> |Dataset|CIFAR-10|CIFAR-100|SVHN|FMNIST|CINIC-10|FEMNIST
> |:-----:|:-----:|:-----:|:-----:|:-----:|:-----:|:---:|
> |FedAvg|76.32% |63.14% | 85.29%| 84.87% |74.62%|76.73%
> |FedFed(Ours)|**82.64%** |**67.32%** | **89.42%**| **88.34%** |**86.39%**|**82.98%**
>
> Table 4. The comparison of our methods on various datasets with the ResNet-18 classifier.
> |Dataset|CIFAR-10|CIFAR-100|SVHN|FMNIST|CINIC-10|FEMNIST
> |:-----:|:-----:|:-----:|:-----:|:-----:|:-----:|:---:|
> |FedAvg|79.35% |67.84% | 88.34%| 86.37% | 78.19%|79.28%
> |FedFed(Ours)|**92.34%**|**69.64%** | **93.21%**| **92.34%** | **90.72%**|**85.74%**
>
> > Q6: Feature distillation idea is proposed in [R1].
>
> **A6:** Thanks for the kind suggestion. In our revision, we will add the following discussion to highlight the difference between our work and previous work [R1] introducing feature distillation.
>
> - Both [R1] and FedFed aim to distil features in the data for better generalization performance. However, the following differences make these two works distinct.
> - 1) [R1] distils features in the representation space, i.e., features extracted by a hidden layer of deep neural networks, while FedFed distil features in the data space.
> - 2) [R1] distils features to guide student models to generate outputs similar to teacher models, while FedFed distils features to share across clients to tackle data heterogeneity.
>
> **Reference**\
> [R1] Romero A, Ballas N, Kahou S E, et al. Fitnets: Hints for thin deep nets. arXiv preprint arXiv:1412.6550.

---

> > ### Comment · Reviewer_wKGY · 2023-08-10
> >
> > Thanks for the response and for running additional experiments.
> >
> > Another suggestion is to provide concrete yet concise examples when providing definitions for performance-robust and performance-sensitive features.
> >
> > Could you please expand on the following: "while FedFed distills features in the data space"? What are features in the data space, and how are they different from the hidden layer features employed in [R1]?

---

> > > ### Author Response · Authors · 2023-08-11
> > > **Response to New Questions from Reviewer wKGY**
> > >
> > > Thanks for your further professional suggestions!
> > >
> > > > Q1: provide concrete yet concise examples when providing definitions for performance-robust and performance-sensitive features.
> > >
> > > **A1:** Following your suggestion, we explicate the definition of performance-sensitive features and performance-robust features from the Information Bottleneck perspective. In brief,
> > >
> > > - At first, we define what a valid partition is (Definition 1);
> > > - Then, following the rules of valid partition, we formalize the two types of features (Definition 2).
> > >
> > > We will elaborate on the definition and explanation in our revision in terms of the following contents.
> > >
> > > *Definition 1. [valid partition] A partition strategy is to partition a variable $X$ into two parts in the same measure space such that $X=X_1 + X_2$. We say a partition strategy is valid if it holds:  i. $H(X, X_1, X_2)=H(X); ii. H(X|X_1, X_2)=0; iii. I(X_1;X_2)=0$, where $H(\cdot)$ denotes the information entropy and $I(\cdot)$ is the mutual information.*
> > >
> > > Definition 1 captures the desirable attributes when we partition the features, which are abstracted to be a variable in general. Intuitively,
> > >
> > > - For i and ii: A valid partition should maintain all information of the original variable $X$ losslessly. That is, neither extra information is introduced nor key information is lost.
> > > - For iii: After determining the measure space, we partition the $X$ to be $X=X_1+X_2$. The mutual information of $X_1$ and $X_2$ is none. Here, $X_1$ and $X_2$ can be symmetric. Or saying, there is no overlap between $X_1$ and $X_2$.
> > >
> > > Built upon the valid partition, we present the definition between performance-sensitive features and performance-robust features, as in Definition 2.
> > >
> > > *Definition 2. Let $X=X_s + X_r$ be a valid partition strategy. We say $X_s$ is performance-sensitive features such that $I(X; Y|X_s)=0$, where $Y$ is the label of $X$. Accordingly, $X_r$ is the performance-robust feature.*
> > >
> > > Intuitively, performance-sensitive features contain all label information, while performance-robust features contain all information about the data except for the label information. That is, the features to be partitioned are either performance-sensitive features or performance-robust features.
> > >
> > > Appreciate again for your kind and constructive suggestion. We believe that our paper will be more clear and readable through your writing-shepherding.
> > >
> > > > Q2: Could you please expand on the following: "while FedFed distils features in the data space"? What are features in the data space, and how are they different from the hidden layer features employed in [R1]?
> > >
> > > **A2:** Thanks for your helpful comments. We hope the following explanation would make the difference clear between FedFed and [R1].
> > > - In FedFed, we distil features in the data space, where we partition raw data $\mathbf{x} \in \mathbb{R}^{d}$ into two parts, i.e., $\mathbf{x}=\mathbf{x}_s+\mathbf{x}_r, \mathbf{x}_s \in \mathbb{R}^{d}, \mathbf{x}_r \in \mathbb{R}^{d}$. Thus, the aim of FedFed is to distil $\mathbf{x}_s$ from $\mathbf{x}$.
> > > - To perform knowledge distillation (i.e., the teacher-student paradigm), [R1] tries to guide student models to generate outputs similar to teacher models using the learned features $\mathbf{f}$. Here, the learned feature is extracted from the neural network $\phi$, i.e., $\mathbf{f}=\phi(\mathbf{x})$.
> > >
> > > Thanks again. We will elaborate on the corresponding explanation in the paper.
> > >
> > > **Reference**
> > > [R1] Romero A, Ballas N, Kahou S E, et al. Fitnets: Hints for thin deep nets. arXiv preprint arXiv:1412.6550.

---

> > > > ### Comment · Reviewer_wKGY · 2023-08-12
> > > >
> > > > So does it mean that FedFed reconstructs x_s from x? Does the paper use the term "distil" for input reconstruction? Otherwise, it is not clear how distillation is working in this context, if it is different than knowledge distillation?
> > > >
> > > > I would be happy to raise the score if this point is resolved.

---

> > > > > ### Author Response · Authors · 2023-08-12
> > > > > **Response to Questions from Reviewer wKGY**
> > > > >
> > > > > **A1:** Thanks for your swift reply, active interaction, and responsible help!
> > > > >
> > > > > Yes, FedFed employs ``distill`` for input reconstruction, i.e.,  reconstructing $\mathbf{x}_s$ from $\mathbf{x}$ and sharing the DP-protected $\mathbf{x}_s$ to tackle data heterogeneity.  We sincerely express our deep and heartfelt appreciation to the reviewer wKGY for the constructive and insightful comments.
> > > > >
> > > > > Really glad to hear the enhanced endorsement of our paper. Your professional suggestions provide immense help to make our paper more clear and concise. We will dedicate our great efforts to improving the next version of our paper. Thanks again！

---

> > > > > > ### Comment · Reviewer_wKGY · 2023-08-12
> > > > > >
> > > > > > Then it is better to call it `input reconstruction`.
> > > > > > Based on your rebuttal and additional experiments, I have increased the score.

---

> > > > > > > ### Author Response · Authors · 2023-08-13
> > > > > > > **Gratitude to wKGY**
> > > > > > >
> > > > > > > We are glad that our responses addressed your questions. We want to express our gratitude once again for upgrading your score!

---

### Official Review · Reviewer_bAmN · 2023-07-06

**Soundness:** 3 good
**Presentation:** 3 good
**Contribution:** 2 fair
**Rating:** 5
**Confidence:** 3

**Summary:**

The paper proposes Federated Feature distillation, a method that addresses the tradeoff between privacy and model performance. It involves extracting performance-robust and performance-sensitive features from local data. The latter is shared among clients after applying differential privacy for privacy preservation. The paper also includes an empirical evaluation demonstrating the effectiveness of the proposed approach.


**Strengths:**

The paper is well-written and easily comprehensible, with clear motivation and contribution.
This paper theoretically shows that the proposed method achieves the same level of privacy with a relatively smaller noise compared to sharing the raw data.
The proposed method can be seamlessly combined with existing universal methods,


**Weaknesses:**

- Additional communication costs:
As sharing globally shared data to all clients, it requires additional communication costs. In F.7 the authors analyze additional communication costs is as same as sending a classifier in approximately 14 communication rounds. But this analysis doesn’t consider the partial participating of federated learning. If we assume the 10% participation rate, sharing the global dataset is equal to the communication costs of 140 communication rounds, which is not small.

- More local iterations on training:
The presence of a globally shared dataset in FedFed results in each client having a significantly larger amount of local training data, K+1 times more than before. It also results in K+1 times local iterations, which can be computationally expensive for edge devices in federated learning, especially when dealing with massively distributed data in realistic settings.
Furthermore, the considerably higher number of local updates in FedFed compared to the approach without FedFed makes it hard to attribute the observed gain in empirical results to a specific factor. It is well-known that increasing local updates can expedite the convergence speed in federated learning. Therefore, the observed gain in empirical results cannot be solely attributed to the FedFed approach, as the increased local iterations inherently provide an advantage. This factor should be taken into consideration when evaluating and comparing the performance of FedFed against other approaches.

- There are lines of work that try to share the client data while preserving the privacy inspired by mixup, such as [Shin et al., 2020] and [Yoon et al., 2021]. These methods are not compared as baselines in evaluation results.


[Shin et al., 2020] MyungJae Shin, Chihoon Hwang, Joongheon Kim, Jihong Park, Mehdi Bennis, and Seong-Lyun Kim. Xor mixup: Privacy-preserving data augmentation for one-shot federated learning. In ICML, 2020.

[Yoon et al., 2021] Tehrim Yoon, Sumin Shin, Sung Ju Hwang, Eunho Yang. Fedmix: Ap- proximation of mixup under mean augmented federated learning. In ICLR, 2021.


**Questions:**

- What is the model's performance when trained solely on the constructed global dataset ($D^s$)?
- Are the models ($w_k^t$) trained in the Feature Distillation phase identical to the global model ($\phi_k^t$) in the local training phase? If they are the same, do the empirical results, in both cases with and without FedFed, initialize the global model in the local training phase with the identical parameters?
- In section F.3, it is observed that sharing the partial feature with DP achieves higher accuracy compared to sharing the raw data with DP, which seems counterintuitive. What is the underlying insight or explanation for this phenomenon?


**Limitations:**

Limitation about storage overhead is stated while potential societal impact has not been addressed.

---

> ### Author Rebuttal · Authors · 2023-08-09
>
> ## Response to Reviewer bAmN:
> > Q1. Additional communication costs: As sharing globally shared data to all clients, it requires additional communication costs. F.7 doesn’t consider the partial participation of federated learning.
>
> **A1:** Thank you for bringing this potentially confusing problem to our attention. We will address it in the upcoming revision by incorporating a detailed explanation and analysis. In FedFed, the constructed dataset is shared only once, leading to extra communication costs. Detailed analysis of communication costs can be found in **Joint Response**.
>
> > Q2: More local iterations on training: The presence of a globally shared dataset in FedFed results in each client having a significantly larger amount of local training data, K+1 times more than before. Increasing local updates can expedite the convergence speed in federated learning. Therefore, the observed gain in empirical results cannot be solely attributed to the FedFed approach, as the increased local iterations inherently provide an advantage. This factor should be taken into consideration when evaluating and comparing the performance of FedFed against other approaches.
>
> **A2:** Thanks for your constructive suggestions. We will highlight the local iterations issue in our revision.
> - The local iterations are doubled in our experiments since we merely sample the same number of samples in each round to perform local training. In our implementation, we expand the batch size (shared and local samples), rather than increase the local iteration, leading to the same number of local iterations as FedAvg without FedFed.
> - The perspective about more local iterations is insightful and consistent with our empirical observations. Accordingly, we will follow the direction to explore more possibilities of FedFed. Note that we also find (only one case) increasing local iterations does not always work (c.f. Table 4 on the main page). We believe it would bring something informative, and we are willing to share it with you.
>
> > Q3: More relative baselines.
>
> **A3:** Following your constructive suggestion, we add more experiments to compare FedFed with other information-sharing strategies [R2][R3]. The results are listed in Table 1. The findings clearly demonstrate that FedFed surpasses advanced information-sharing methods in terms of performance. We will include these results and provide a comprehensive discussion in our revision.
>
> Table 1. Comparison with information-sharing methods under $\alpha=0.1, E=1, K=100$ over CIFAR-10.
> |FedAvg [R1]|Xor mixup [R2]|FedMix [R3]|FedAvg with FedFest(Ours)|
> |:------:|:--------:|:-------:|:-----------:|
> |49.72%|76.18%|78.35%|**84.06%**|
>
> > Q4. What is the model's performance when trained solely on the constructed global dataset $(D)^s$
>
> **A4:** Thanks for the inspiring question! Following the interesting direction, we train models merely on the constructed global dataset and report the results as follows in Table 2. We can see that the model's performance is bad. We guess the distribution shift between the constructed data and the natural data causes bad performance. However, we believe that training models merely on noisy data is a promising direction and we will explore it in our future work.
>
> Table 2. The difference between training in FedFed manner and solely on shared data.
> |  	| CIFAR-10 	|
> |:---:|:---:|
> | FedFed(Ours) | 92.34% |
> | Solely on constructed global dataset 	| 27.68% 	|
>
> > Q5. Are the models ($\omega^t_k$) trained in the Feature Distillation phase identical to the global model ($\phi_k^t$) in the local training phase? If they are the same, do the empirical results, in both cases with and without FedFed, initialize the global model in the local training phase with identical parameters?
>
> **A5:** Thanks for your inspiring questions.
> - In our experiments, the model parameters $\omega^t_k$ and $\phi_k^t$ are not the same. The $\omega^t_k$ is used to extract performance-sensitive features while $\phi_k^t$ is the local classifier like other FL methods.
> - We agree with your point that it is promising to initialize the global model with models trained for feature generation. Accordingly, we use the same model for the mentioned two models and initialize the global model with local models trained for feature generation.
> The results are listed in Table 3. The results show that employing the same model will further promote the performance and speed up the convergence rate of FedFed. Thanks again for the inspiring and helpful questions!
>
> Table 3. Comparison between the initialization of $\phi_k^t$  with/without $\omega^t_k$.
> |  	| Accuracy | Round |
> |---|---|---|
> | FedFed(the initialization of $\phi_k^t$  without $\omega^t_k$) | 92.34% | 39 |
> | Inspired experiment(the initialization of $\phi_k^t$  with $\omega^t_k$) | 92.94% | 31 |
>
> > Q.6 In section F.3, it is observed that sharing the partial feature with DP achieves higher accuracy compared to sharing the raw data with DP, which seems counterintuitive. What is the underlying insight or explanation for this phenomenon?
>
> **A6:** Thanks for pointing out the potentially confusing problem. Models trained over raw data with noise perform worse than those trained over partial data since the privacy budget for these two cases is the same. Namely, to achieve the same privacy budget, raw data are injected with noise on a larger scale than partial data. We will add the explanation to our revision to fix the problem.
>
> **Reference**\
> [R1] McMahan B, Moore E, Ramage D, et al. Communication-efficient learning of deep networks from decentralized data. In AISTAS,2017.\
> [R2] MyungJae Shin, Chihoon Hwang, Joongheon Kim, Jihong Park, Mehdi Bennis, and Seong-Lyun Kim. Xor mixup: Privacy-preserving data augmentation for one-shot federated learning. In ICML, 2020.\
> [R3] Tehrim Yoon, Sumin Shin, Sung Ju Hwang, Eunho Yang. Fedmix: Ap- proximation of mixup under mean augmented federated learning. In ICLR, 2021.

---

> > ### Comment · Reviewer_bAmN · 2023-08-14
> >
> > Thanks the authors for the response. Although the proposed method requires considerable extra communication cost in some practical settings, other my concerns have been addressed in the rebuttal. I will keep my original score as is.

---

> > > ### Author Response · Authors · 2023-08-14
> > > **Response to Reviewer bAmN**
> > >
> > > We are glad that we have addressed your concerns. Thanks again for your valuable comments.

---

### Official Review · Reviewer_LGae · 2023-07-10

**Soundness:** 2 fair
**Presentation:** 3 good
**Contribution:** 3 good
**Rating:** 6
**Confidence:** 3

**Summary:**

The paper introduces a federated learning framework (FedFed) to tackle data heterogeneity by utilizing an information-sharing approach. The method partitions data into performance-sensitive features and performance-robust features, based on their contribution to model performance. The performance-sensitive features are shared globally to mitigate data heterogeneity, while the performance-robust features are kept locally. The method employs DP to protect performance-sensitive features before sending them to the server.

**Strengths:**

+ Improving how to handle data heterogeneity in federated learning is a timely and important problem.
+ The idea of tackling data heterogeneity from the Information Bottleneck perspective is interesting.
+ The solution appears to significantly boost training performance of various FL algorithms and (in most cases) reduces the required number of communication rounds, while maintaining privacy.

**Weaknesses:**

- Overall the practicality of the method is unclear.
- The evaluation only considers a single task and uses synthetic non-IID data.
- The evaluation should include additional baselines.

**Questions:**

This is an interesting approach but it's unclear how practical it is.
The method appears to have several overheads that are not quantified: the computation overhead to generate the performance-robust/sensitive features and to train the local classifier, communication overhead to collect protected features from clients to construct a global dataset and send the dataset back to the clients, and storage overhead to save the models and the private and public datasets on each client. This might make the method not feasible for edge devices with limited capabilities.

The experimental results focus on image classification task, with no experiments with other tasks. Also, the experiments rely on extreme non-IID partitions ( α=0.1 and α=0.05 ). It would be useful to test the method with different levels of heterogeneity (e.g., α=0.5 and α=1) and with datasets with natural non-IID partitions like Stack Overflow or Reddit.

The evaluation should consider other baselines. It is good that FedFed can be applied on top of FL algorithms. But other extensions to those algorithms should be considered as a baseline too: an obvious one was mentioned by the authors, where the full data + random noise is shared. This baseline will tell us how much better FedFed is than the naive solution. The authors mentioned that this will result in an accuracy loss, but, without evidence to support this.

The hyperparameters used in the experiments are not clear, and we don't know if they were tuned to get the best performance for each algorithm or not. The client selection hyperparameter is not mentioned. It is not clear how many clients will be sampled from the $K$ clients and what the selection method is.

The method is described as an extension on top of FL algorithms, but the paper does not explain how the extension can be used with other algorithms. In fact, algorithm 1 depicts FedFed as a new FL algorithm and not an extension.

It is not clear where $\mathcal{D}^s$ is used, which is why the entire algorithm is proposed! Note that there is a typo in eq. 8.

The preliminaries section should introduce information bottleneck.

Other questions that come up:

1. FedFed uses eq. 8 in the local update; but how does it extend SCAFFOLD, FedProx, and FedNova since they modify the local update as well?
2. The reported FedNova performance in table 1 does not seem to agree with the performance reported in their paper, where they always outperformed FedAvg; why is that?
3. Why the local epochs E was set to 1 and 5? Why 5 local epochs is not used with the experiment with 100 clients?
4. How many clients were sampled at every round? What is the value of the client selection hyperparameter?

*I acknowledge I have read the authors' rebuttal and answers to my questions.*

**Limitations:**

* The paper has discussed the storage limitation but didn't clarify how much extra storage overhead was required and the size of the shared global datasets.
* The paper didn't address how much per-round extra communication overhead the method required; although the approach seems to reduce the number of communication rounds significantly in most cases, there were some cases (table 4) that required more communication rounds compared to the other baselines. Moreover, communicating $x_p$ and $\theta$ adds extra communication overhead. Clearly, FedFed gains are not a free lunch, and these limitations must be considered, discussed, and be part of the evaluation.

---

> ### Author Rebuttal · Authors · 2023-08-10
>
> ## Response to Reviewer LGae:
> > Q1: Some overheads are not quantified
>
> **A1:** Thank you for your insightful comments. Accordingly, we add analysis about overheads in **Joint Response**.
>
> > Q2: Test with different levels of heterogeneity and datasets with natural non-IID partitions.
>
> **A2:** Thanks for your detailed suggestions.
> - Heterogeneity levels. We conduct additional experiments on $\alpha=0.5$ and $\alpha=1.0$, $\beta$ is the sampling rate. The results are reported in Tables 1-4.
>
> Table 1. Accuracy of $\alpha=0.5$ with 10 clients, $\beta=$50%.
> || CIFAR-10|CIFAR-100|SVHN|FMNIST|
> |--|--|--|--|--|
> | FedAvg | 84.56% | 68.78%| 90.21%| 88.78%|
> | FedFed(Ours)| **93.37%** | **70.91%** | **93.71%** | **94.01%**|
>
> Table 2. Accuracy of $\alpha=1.0$ with 10 clients,$\beta=$50%.
> | | CIFAR-10| CIFAR-100| SVHN| FMNIST|
> |--|--|--|--|--|
> |FedAvg| 87.18%| 69.12%| 92.37%|90.98%|
> |FedFed(Ours)|**94.07%**|**71.01%**|**93.98%**|**95.12%**|
>
> Table 3. Accuracy of $\alpha=0.5$ with 100 clients,$\beta=$10%.
> ||CIFAR-10|CIFAR-100|SVHN|FMNIST|
> |--|--|--|--|--|
> | FedAvg       | 59.78%     | 54.56%     | 90.31%     | 91.78%     |
> | FedFed(Ours) | **89.19%** | **63.78%** | **92.09%** | **93.13%** |
>
> Table 4. Accuracy of $\alpha=1.0$ with 100 clients, $\beta=$10%.
> || CIFAR-10|CIFAR-100|SVHN|FMNIST|
> |--|--|--|--|--|
> |FedAvg|63.19%|57.19%|91.01%|92.98%|
> |FedFed(Ours)|**90.31%**|**66.89%**|**93.01%**|**93.81%**|
>
> We can see that FedFed significantly improves performance.
> - Realistic Non-IID Partitions. We have verified the effectiveness of FedFed on the FEMNIST dataset [R4], which has more realistic non-IID partitions. The results are reported in Table 5. We can see that FedFed performs well on FEMNIST. We did not conduct experiments on the Stack Overflow and Reddit datasets as mentioned, as these datasets are typically used for the next character prediction, which differs from the classification task studied in our work. Inspired by your valuable comments, we will explore FedFed under more complex scenarios.
>
> Table 5. The results on FEMNIST
> ||FEMNIST|
> |--|--|
> |FedAvg|79.28%|
> |FedFed(Ours)|85.74%|
>
>  We will add these results to our updated revision.
> > Q3: Add baseline: the full data + random noise is shared
>
> **A3:** We apologize for omitting important information. We will highlight the results (reported in Appendix F.3) on the main page. These results show that simply applying noise to the data causes performance degradation. Other comparisons with existing sharing strategies can be found in the table provided in **A1 of response to Reviewer xchj**.
>
> > Q4. The client selection hyperparameter is not mentioned and what the selection method is.
>
> **A4:** Thank you for pointing out the confusing parts. We list hyperparameters in Table 7 in Appendix B.1 (i.e., Implementation Details). We apologise for missing how to determine the hyperparameter $\rho$. In FedFed, each client retains a small portion of data that is not used for training. After several rounds of training, these reserved data are utilized to select the hyperparameter $\rho$. We set the sampling rate to be 10% for 100 clients and 50% for 10 clients. We randomly select clients from the client list. Motivated by your comments, we conduct experiments with different sample rates to verify the efficacy of FedFed. Results are given in Tables 1-11 in **Joint Response**.
>
> > Q5. The paper does not explain how the extension can be used with other algorithms.
>
> **A5:** Thank you for your detailed comments. Algorithm 1 presents how we can obtain the globally shared dataset $D^s$. In Appendix B.2 (Pseudo-code and Explanation), we demonstrate how to implement FedFed with existing methods as a plug-in module. We highlight the modification of the original algorithms with only two lines. We will elaborate on the contents in our updated revision.
>
> > Q6. The reported FedNova performance in Table 1 does not seem to agree with the performance reported in their paper; why is that?
>
> **A6:** In FedNova, the main results were reported using $K=16$ and $\alpha=0.1$, while the results using $K=100$ clients and $\alpha=0.05$ were omitted. Our experiments are consistent with those reported in [R1].
>
> > Q7. Why 5 local epochs is not used in the experiment with 100 clients?
>
> **A7:** We followed the settings used in previous works [R2][R3]. In response to your valuable comments, we conducted additional experiments using 100 clients with 5 local epochs. The results reported in Table 6 demonstrate that FedFed can provide a significant improvement. We will include these results in our updated revision.
>
>  Table 6 The accuracy of $\alpha=0.1, E = 5, K= 100$  over various datasets.
> |Method|CIFAR-10 |FMNIST|SVHN|CIFAR-100|
> |--|--|--|--|--|
> |FedAvg|60.78%|91.21%|90.31%|51.69%|
> |FedFed(Ours)|**88.21%**|**92.19%**|**92.78%**|**62.77%**|
>
> > Q8. there is a typo in eq. 8.
>
> **A8:** We have fixed the typo in the revision. Thanks for your attentive review!
>
> > Q9. The preliminaries section should introduce an information bottleneck.
>
> **A9:** Thanks for your valuable suggestions. We will add the necessary introduction in the revised version
>
> > Q10. How many clients were sampled at every round?
>
> **A10:** We set the sample rate to be $10\%$ for 100 clients and $50\%$ for 10 clients. More details of experiments can be found in **Joint Response**.
>
> > Q11: limitations must be considered, discussed, and be part of the evaluation.
>
> **A11:** Please see **Joint Response**. Following your constructive suggestion, we will highlight the overhead introduced by FedFed on the main page.
>
> **Reference**\
> [R1] Li et al. Federated learning on non-iid data silos: An experimental study. In ICDE, 2022.\
> [R2] Tang et al. Virtual homogeneity learning: Defending against data heterogeneity in federated learning. In ICML, 2022.\
> [R3] Hsu et al. Measuring the effects of non-identical data distribution for federated visual classification. arXiv:1909.06335.\
> [R4] Caldas et al. Leaf: A benchmark for federated settings. arXiv:1812.01097.

---

> > ### Comment · Reviewer_LGae · 2023-08-20
> >
> > Thanks to authors for their comments. As several concerns have been addressed (comp. and comm. overheads), and more corroboration of results, I will improve my score.

---

> > > ### Author Response · Authors · 2023-08-20
> > > **Response to LGae**
> > >
> > > We are happy that we have addressed your concerns. Thank you for the effort you put into improving the quality of our paper. Thanks again for raising the score.

---

### Official Review · Reviewer_xchj · 2023-07-13

**Soundness:** 3 good
**Presentation:** 3 good
**Contribution:** 3 good
**Rating:** 5
**Confidence:** 2

**Summary:**

Authors propose method (Federated Feature distillation) to share partial features in the data to tackle data heterogeneity, while the privacy issue is not compromised too much. FedFed partitions data into performance-sensitive features and performance-robust features. The performance-sensitive features are globally shared to mitigate data heterogeneity, while the performance-robust features are kept for local training.

**Strengths:**

1. Authors claim to propose a new prospect on alleviating date heterogeneity in Federated Learning: sharing partial features. It is a new high-level idea to solve heterogeneity in FL.
2. To achieve the idea, authors further propose a method (FedFed). More specifically, it proposed to partition data into performance-sensitive features and performance-robust features is reasonable. The performance-sensitive features are globally shared to mitigate data heterogeneity, while the performance-robust features are kept locally. This idea is resonable.
3. It outperfroms baselines



**Weaknesses:**

1. Insufficient baselines. I wonder how the choices of baselines are made? FedAvg, FedProx, FedNova and SCAFFOLD are early FL models. The latest among them is proposed in 2020. Why not adopt more recent FL models? For instance, FedGen (ICML 2021)[1]. Besides, I believe there are more FL models in recent years.

[1] Zhu et al., Data-Free Knowledge Distillation for Heterogeneous Federated Learning. ICML 2021.



**Questions:**

See weakness

**Limitations:**

Yes

---

> ### Author Rebuttal · Authors · 2023-08-09
>
> ## Response to Reviewer xchj:
> > Q1: Adopt recent baselines, e.g., FedGen (ICML 2021)[R6]
>
> **A1:**  Following your constructive suggestion, we have compared FedFed with open-source mainstream information-sharing-based works that aim to mitigate data heterogeneity.
> The results are reported in Table 1. We can see that FedFed outperforms existing methods. We will add the results and discussion to our updated revision.
>
> Table 1. Comparison with baselines under $\alpha=0.1, E=1, K=100$ over CIFAR-10.
>
> |FedAvg [R1]|FedAvg with CCVR [R2]|FD+FAug [R3]|FedMD [R4] |FedDF [R5]|FedGen[R6]|FedProto [R7]|FedFTG [R8]|FedAvg with FedFest(Ours)|
> |:------:|:--------:|:-------:|:-----------:|:--:|:--:|:--:|:-:|:-:|
> |49.72%| 59.19%|63.54%|67.32%|73.41%|76.45%|77.08%|80.73%|**84.06%**|
>
> **Reference**\
> [R1] McMahan B, Moore E, Ramage D, et al. Communication-efficient learning of deep networks from decentralized data. In AISTAS,2017.\
> [R2] Luo M, Chen F, Hu D, et al. No fear of heterogeneity: Classifier calibration for federated learning with non-iid data. In NeurIPS, 2021.\
> [R3] Jeong E, Oh S, Kim H, et al. Communication-efficient on-device machine learning: Federated distillation and augmentation under non-iid private data. arXiv preprint arXiv:1811.11479.\
> [R4] Li D, Wang J. Fedmd: Heterogenous federated learning via model distillation. arXiv preprint arXiv:1910.03581.\
> [R5] Lin T, Kong L, Stich S U, et al. Ensemble distillation for robust model fusion in federated learning. In NeurIPS, 2020.\
> [R6] Zhu Z, Hong J, Zhou J. Data-free knowledge distillation for heterogeneous federated learning. In ICML, 2021.\
> [R7] Tan Y, Long G, Liu L, et al. Fedproto: Federated prototype learning across heterogeneous clients. In AAAI, 2023.\
> [R8] Zhang L, Shen L, Ding L, et al. Fine-tuning global model via data-free knowledge distillation for non-iid federated learning. In CVPR, 2022.

---

### Author Rebuttal · Authors · 2023-08-09

## Response to All Reviewers:
We sincerely appreciate all reviewers' great efforts on review and comments on our work. We especially thank the nice words:
- timely and important problem (Reviewer #LGae)
- new and interesting high-level idea (Reviewer #xchj & #LGae) and  reasonable and simple method (Reviewer #xchj & #wKGY)
- significantly boosts training performance, reduces communication rounds, and maintains privacy (Reviewer #LGae )
- empirical evaluation (Reviewer #ezSw & #xchj & #LGae) and theoretical justification (Reviewer #bAmN & #wKGY)
- seamless combination or easy adoption in further research (Reviewer #xchj & #LGae & #bAmN & wKGY & ezSw)

Due to limited space, we extract a similar question and answer it in response to your valuable comments.

### Joint Response
> Various overheads (Reviewer #LGae & #bAmN) introduced by FedFed:

Thanks for the comments. According to the empirical analyses given in Appendix F.7, we can see that the communication and computation overheads are relatively mild:

#### 1. Computation Overhead
-  The computation overhead, introduced by FedFed to generate performance-robust/sensitive features, is less than 10% of the overall training computation for the local classifier.

#### 2. Communication Overhead
- Consider $K$ clients in the FL system. Let $m$ be the size of a single local model. The size of local private data is $\|D_k\| = a$. Then, the ratio of the entire dataset to the model parameter  is $\gamma$ as in Appendix F.7,
$$ \gamma = \frac{aK}{m}\approx14,$$

- The extra communication cost for a single client is $$(m+m)*T_d+a+aK$$ where $(m+m)*T_d$ denotes the cost of download/upload models for $T_d$ rounds in Algorithm 1. Here, the $a$ denotes the performance-sensitive features sent by each client, and $aK$ is the data received by each client from the globally shared dataset.

- In the general process of FL, the overall communication costs are $(m+m)T_r\beta$, where $\beta$ is the sampling rate of a client. Therefore, the ratio of the extra communication overhead to the general FL process is:
$$\frac{(m+m)\cdot T_d+a(K+1)}{(m+m)\cdot T_r\cdot \beta} = \frac{T_d}{T_r\cdot\beta}+\frac{a(K+1)}{(m+m)\cdot T_r\cdot\beta} = \frac{T_d}{T_r\cdot\beta}+\frac{\gamma}{2\cdot T_r \cdot\beta}+\frac{\gamma}{2\cdot K\cdot T_r \cdot \beta}$$

Here, we detail two examples in our experiments.
- For $K=10, T_d=15$, $T_r=1000$, and $\beta=$50%, the extra communication costs are approximately 4.54%.
- When $K=100$, $T_d=15$, $T_r=1000$, and $\beta=$10%, the extra communication costs are approximately 22.07%.

#### 3. Storage Overhead
FedFed offers three trade-off strategies regarding communication bandwidth and storage.
- (i) One-time download: Local clients download the globally shared dataset once. Globally shared dataset costs approximately $14 \times$ storage of classifier model.
- (ii) Partial download: A small portion of the globally shared dataset is selected and downloaded. This strategy incurs approximately $1.5\times$ communication cost compared to the previous strategy, while the storage required by the clients is the same as that of local private data.
- (iii) Intermittent download: A small set of globally shared dataset is downloaded after every $Z$ rounds. This approach reduces the communication overhead to $\frac{1}{Z}$ of that of strategy (ii) while maintaining the storage overhead at the size of the local data.

> Experiments for different sampling rates (Reviewer #LGae):

Thanks for the comment. Inspired by Reviewer #LGae, we conduct experiments to investigate the impact of different sampling rates on performance and convergence speed, the results can be found in Tables 1-11.

Table 1. The accuracy of  $\alpha=0.1, E = 1, K= 10$ over CIFAR-10.
|Sampling rate|10% |20%|30%|40%|50%|
|--|--|--|--|--|--|
|Accuracy|88.58%|89.34%|91.66%|91.94%|92.34%
|Round|102|73|52|48|39|

Table 2. The accuracy of $\alpha=0.05, E = 1, K= 10$ over CIFAR-10.
|Sampling rate|10% |20%|30%|40%|50%|
|--|--|--|--|--|--|
|Accuracy|82.77% |84.47% | 86.82%| 88.98% | 90.02%
|Round| 97| 74| 62| 57| 50|44

 Table 3. The accuracy of $\alpha=0.1, E = 1, K= 100$ over CIFAR-10.
|Sampling rate|5% |10%|20%|40%|60%|
|--|--|--|--|--|--|
|Accuracy| 83.67%|84.06%|87.98%|89.17%|89.62%
|Round|182|163|90|70|61|

 Table 4. The accuracy of $\alpha=0.1, E = 1, K= 10$ over CIFAR-100.
|Sampling rate|10% |20%|30%|40%|50%|
|--|--|--|--|--|--|
|Accuracy|63.34%|64.29% | 66.47%|67.32% |69.64%|
|Round| 405| 389 | 331| 291 | 283|

 Table 5. The accuracy of under $\alpha=0.05, E = 1, K= 10$ over CIFAR-100.
|Sampling rate|10% |20%|30%|40%|50%|
|--|--|--|--|--|--|
|Accuracy|63.77%|65.12% | 66.02%|66.82% |68.49%|
|Round| 302| 234 | 198| 156 | 137|

 Table 6. The accuracy of $\alpha=0.1, E = 1, K= 100$ over CIFAR-100.
|Sampling rate|5% |10%|20%|40%|60%|
|--|--|--|--|--|--|
|Accuracy| 57.32%|60.58%|64.34%|66.53%|68.71%
|Round|534|448|401|382|298|

Table 7. The accuracy of $\alpha=0.1, E = 1, K= 10$ over FMNIST.
|Sampling rate|10% |20%|30%|40%|50%|
|--|--|---|--|--|--|
|Accuracy|88.36%|90.34% | 91.78%|92.01% |92.34%|
|Round| 60| 35 | 28| 21 | 14|

 Table 8. The accuracy of $\alpha=0.05, E = 1, K= 10$ over FMNIST.
|Sampling rate|10% |20%|30%|40%|50%|
|--|--|--|--|--|--|
|Accuracy|87.34%|88.78% | 90.11%|90.34% |90.69%|
|Round| 43| 32 | 30| 22 | 16|

 Table 9. The accuracy of $\alpha=0.1, E = 1, K= 100$ over FMNIST.
|Sampling rate|5% |10%|20%|40%|60%|
|--|--|--|--|--|--|
|Accuracy| 90.87%|92.71%|92.88%|93.51%|93.69%
|Round|287|243|213|157|104|

Table 10. The accuracy of $\alpha=0.1, E = 1, K= 10$ over SVHN.
|Sampling rate|10% |20%|30%|40%|50%|
|--|--|--|--|--|--|
|Accuracy|89.74%|91.23% | 92.08%|92.82% |93.21%|
|Round| 273| 200|188| 143 | 105|

 Table 11. The accuracy of $\alpha=0.1, E = 1, K= 100$  over SVHN.
|Sampling rate|5% |10%|20%|40%|60%|
|--|--|---|--|--|--|
|Accuracy| 89.37%|91.04%|92.88%|93.51%|93.69%
|Round|803|763|701|637|541|

---

### Decision · Program_Chairs · 2023-09-21

**Decision:**

Accept (poster)

**Comment:**

This paper presents a novel approach named federated feature distillation (FedFed) to deal with the data heterogeneity issue in federated learning. Reviewers agreed that this paper investigates a timely and important problem, the proposed method is very novel, and the experiments are extensive and convincing. The authors' rebuttal has also successfully addressed the concerns from reviewers, such as additional baselines and overhead analysis. The authors are highly encouraged to incorporate the suggestions from reviewers to the final version of this paper.